# Terminal uridyltransferase 7 regulates TLR4-triggered inflammation by controlling Regnase-1 mRNA uridylation and degradation

Chia-Ching Lin [1], Yi-Ru Shen[1], Chi-Chih Chang[1], Xiang-Yi Guo[1], Yun-Yun Young[1], Ting-Yu Lai[1], I-Shing Yu [2], Chih-Yuan Lee[1,3,4], Tsung-Hsien Chuang[5], Hsin-Yue Tsai[1,4] & Li-Chung Hsu [1,4 ✉]

Different levels of regulatory mechanisms, including posttranscriptional regulation, are needed to elaborately regulate inflammatory responses to prevent harmful effects. Terminal uridyltransferase 7 (TUT7) controls RNA stability by adding uridines to its 3′ ends, but its function in innate immune response remains obscure. Here we reveal that TLR4 activation induces TUT7, which in turn selectively regulates the production of a subset of cytokines, including Interleukin 6 (IL-6). TUT7 regulates IL-6 expression by controlling ribonuclease Regnase-1 mRNA (encoded by *Zc3h12a* gene) stability. Mechanistically, TLR4 activation causes TUT7 to bind directly to the stem-loop structure on *Zc3h12a* 3′-UTR, thereby promotes *Zc3h12a* uridylation and degradation. *Zc3h12a* from LPS-treated TUT7-sufficient macrophages possesses increased oligo-uridylated ends with shorter poly(A) tails, whereas oligo-uridylated *Zc3h12a* is significantly reduced in *Tut7⁻/⁻* cells after TLR4 activation. Together, our findings reveal the functional role of TUT7 in sculpting TLR4-driven responses by modulating mRNA stability of a selected set of inflammatory mediators.

[1] Institute of Molecular Medicine, National Taiwan University, Taipei, Taiwan. [2] Laboratory Animal Center, College of Medicine, National Taiwan University, Taipei, Taiwan. [3] Department of Surgery, National Taiwan University Hospital, Taipei, Taiwan. [4] Center of Precision Medicine, College of Medicine, National Taiwan University, Taipei, Taiwan. [5] Immunology Research Center, National Health Research Institutes, Miaoli County, Taiwan. ✉email: lichunghsu@ntu.edu.tw

nflammatory cytokines and chemokines are mediators of inflammation in the innate arm of immunity. Innate cells produce inflammatory cytokines and chemokines rapidly upon infection or tissue injury through an array of pattern-recognition receptors (PRRs)[1]. Inappropriate expression of inflammatory cytokines dysregulates inflammation, causing numerous disorders, including inflammatory and autoimmune diseases. Therefore, orchestrating the expression of inflammatory cytokines for an effective inflammatory response without harmful immunopathology is an important issue.

Recognition of conserved pathogen-associated molecular patterns (PAMPs) on microorganisms and damage-associated molecular patterns (DAMPs) released from stressed or injured cells by PRRs initiates pro-inflammatory cytokine responses and coordinates adaptive immune responses[2]. Amongst all PRR families, Toll-like receptors (TLRs) are the most well-characterized and they play an important role in innate immune responses[2]. TLR4 recognizes lipopolysaccharides (LPS) on Gram-negative bacteria[3]. Upon engagement with its ligand, TLR4 recruits Myeloid differentiated primary response gene-88 (MyD88) and Toll/IL-1 receptor domain-containing adapter-inducing IFNβ (TRIF) to activate inhibitor of transcription factors NF-κB kinase (IKK)/NF-κB and TANK-binding kinase 1 (TBK1)/interferon regulatory factor 3 (IRF3) for the production of pro-inflammatory cytokines/chemokines and type I interferons, respectively[4,5]. Inflammatory responses induced by TLRs or other PRRs are controlled by layers of regulatory mechanisms including: modulation of signaling transduction, posttranscriptional regulation (PTR), and posttranslational modification (PTM)[6,7]. PTR which includes RNA splicing, editing, and decay, is a quick and effective way to control the quantity of mRNA[2,8–10]. The role of PTR in regulation of PRR-triggered inflammatory response has come to light as of late.

Many PRR-triggered cytokine mRNAs have short half-lives which allows rapid control of cytokine production[6]. Cytokine mRNAs are subject to various regulatory cis-elements in their 3′-UTRs, such as microRNA (miRNA) target sites, AU-rich elements (AREs), and stem-loop structures. Many RNA-binding proteins (RBPs), including Regnase-1[11], Arid5a[12], Tristetraprolin (TTP)[13], and AU-rich element RNA-binding protein 1 (AUF-1)[14], specifically bind regulatory RNA cis-elements to control the stability of cytokine mRNAs. TTP and AUF1 recognize AREs in the 3′-UTR of numerous cytokines, including tumor necrosis factor (TNF) and IL-6, to initiate their decay[13,14]. The ribonuclease (RNase) Regnase-1 (also called Mcpip1 encoded by the Zc3h12a gene) modulates IL-6 mRNA stability when macrophages are activated by LPS or IL-1β[15]. Mechanistic studies revealed that Regnase-1 targets the stem-loop structure on Il6 3′-UTR and cleaves Il6 mRNA through cooperation with RNA helicase protein, UPF1[11,16]. On the other hand, stimulation with LPS and cytokines also enhances the expression of Arid5a. The presence of Arid5a impedes Regnase-1 binding to Il6 3′-UTR thereby stabilizes Il6 mRNA[12]. Therefore, it appears that regulation of spatiotemporal expression of RBPs can fine tune the quantity of cytokine mRNAs during inflammation.

Recent studies show that modification of RNAs on their 3′ ends by non-templated nucleotide addition is an evolutionary conserved mechanism for the control of RNA stability and its fate[17]. Among the known template-independent RNA 3′ terminal modifications, polyadenylation and uridylation are the two most studied mechanisms[18]. Uridylation is catalyzed by terminal uridyltransferases (TUTs). TUTs belong to a family of non-canonical poly(A) polymerases. Their function in RNA processing is evolutionarily conserved from *Schizosaccharomyces pombe* to human[19]. TUT4 (also known as ZCCHC11) and TUT7 (also named ZCCHC6) are primarily responsible for cytoplasmic 3′

uridylation[20,21] of various RNAs, including precursor microRNAs (pre-miRNAs)[22,23], mature miRNAs[24], histone mRNAs[25,26], cellular mRNAs[20], noncoding RNAs[27], and viral RNAs[28] in mammalian cells. Catalytic activities of TUT4/7 can mono-uridylate or oligo-uridylate pre-miRNA resulting in biogenesis or degradation of miRNAs, respectively[21,23]. TUT4/7 are also known to trigger oligo-uridylation of poly(A)-tail-lacking histone mRNA at the end of S phase of a cell cycle to enhance its degradation[25,26,29]. Recently, TAIL-seq analysis revealed that TUT4 and TUT7 uridylate mRNAs with short poly(A) tails, leading to 5′-to-3′ or 3′-to-5′ mRNA degradation by recruiting deadenylases, decapping enzymes, and exonucleases[20,30,31]. These studies indicate that oligo-uridylation is associated with RNA degradation, but how TUT4/7 uridylate RNAs remains unknown.

TUTs are recently shown to modulate inflammatory responses in mammalian cells. TUT4 uridylates and degrades IL-6-targeting miR-26a/b, leading to control of the quantity of Il6 mRNA in A549 cells following TNF treatment[24]. TUT7 is also known to modulate miR-26b uridylation and stability in IL-1β-stimulated chondrocytes through which positively regulates IL-6 expression[32]. A recent report revealed that TUT7 impedes the expression of a specific set of pro-inflammatory cytokines (including IL-6) in mice after challenge with *Streptococcus pneumonia*[33]. It is thus evident from in vitro and in vivo studies that TUT7 modulates inflammatory response. However, it remains an interesting question whether TUT7 can modulate inflammation by direct targeting mRNAs.

In this study, we reveal that TLR4 activation in macrophages upregulates TUT7 expression and TUT7 selectively regulates the expression of a subset of inflammatory cytokines. Both in vitro and in vivo studies show that TUT7 is critical for IL-6 and IL-12p40 upon LPS challenge and it is dependent on its nucleotidyltransferase activity. We further demonstrate that TUT7 directly controls Zc3h12a mRNA decay by uridylating its 3′ end, which subsequently prevents Regnase-1 from degrading Il6. Interestingly, Zc3h12a isolated from LPS-treated TUT7-sufficient bone marrow-derived macrophages (BMDMs) contains increased oligo-uridines (≥2 U) with short poly(A) tails at its 3′ end, whereas reduced oligo-uridylation is found in Zc3h12a transcripts from TUT7-deficient cells after the same treatment. Together, our results demonstrate that TUT7 functions as a regulator in TLR4-driven inflammatory responses by mediating uridylation of and thus destabilizing the mRNAs of inflammatory mediators including Zc3h12a.

## Results

**TUT7 regulates cytokine production in macrophage response to LPS**. TUTs possess the ability to uridylate small RNAs and/or mRNAs, thereby regulates their stability[17,21,29]. Transcriptome analysis of GEO Datasets (GDS5623) obtained from the National Center for Biotechnology Information shows that TUT7 is upregulated in both BMDMs and RAW 264.7 macrophages upon LPS stimulation (https://www.ncbi.nlm.nih.gov/sites/GDSbrowser?acc=GDS5623). We discovered that LPS treatment dramatically increased TUT7 expression in BMDMs (Fig. 1a). LPS-induced TUT7 expression required the activities of IKK and p38 Mitogen-activated protein kinase (MAPK), whereas suppression of JNK and ERK MAPKs had no effect on TUT7 expression (Supplementary Fig. 1a). In addition, other TLR agonizts, including Pam$_3$CSK$_4$ (TLR1/2), poly(I:C) (TLR3), R848 (TLR7/8) and CpG-1826 (TLR9) induced moderate levels of TUT7 expression (Supplementary Fig. 1b). IKK, but not JNK, activation was required for Pam$_3$CSK$_4$- and R848-triggered TUT7 expression (Supplementary Fig. 1c).

To assess the role of TUT7 in TLR4-driven inflammatory response, we knocked down its expression in murine RAW 264.7

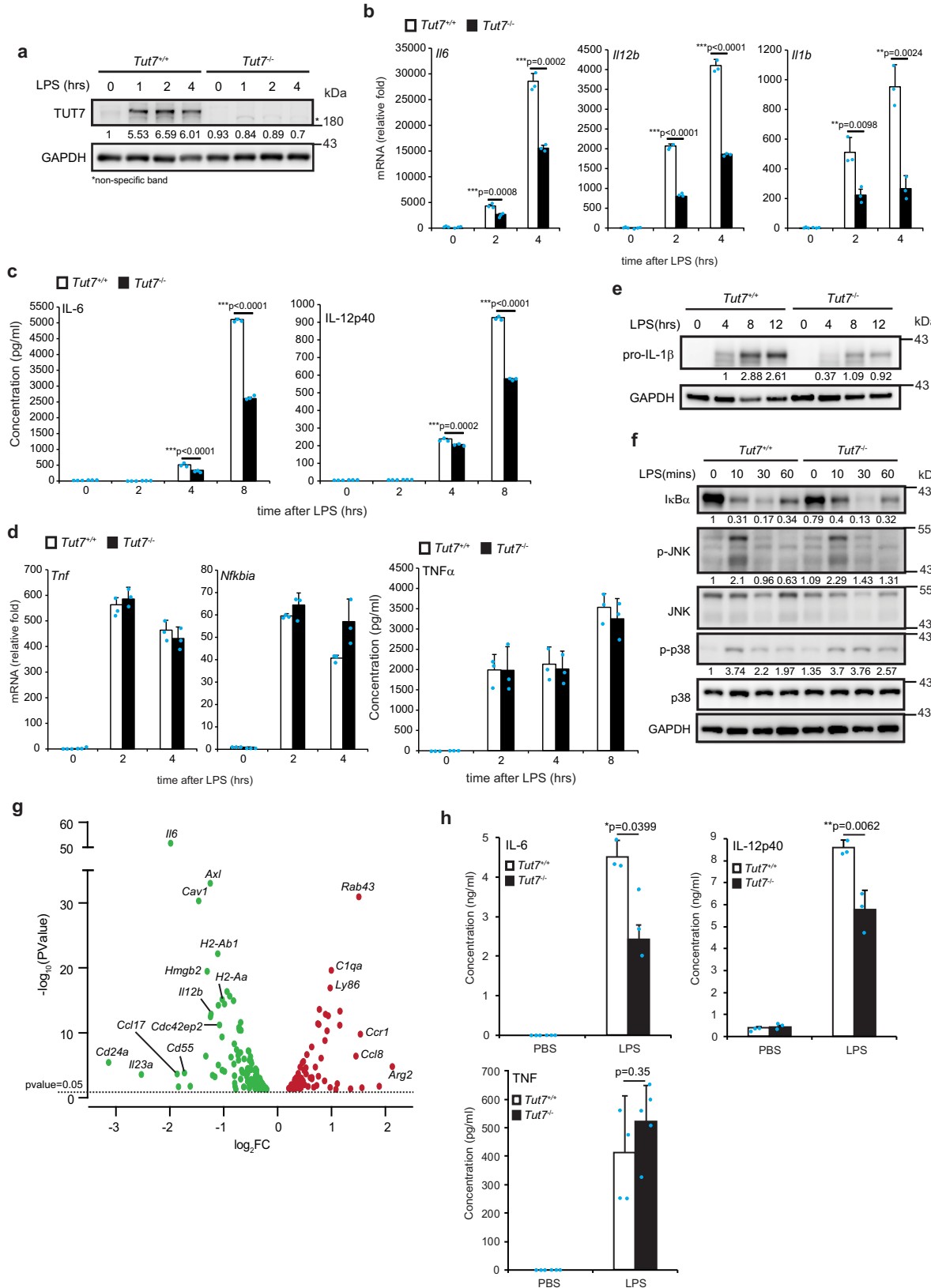

macrophages by lentivirus-mediated shRNA transduction. Silencing *Tut7* in RAW 264.7 macrophages significantly decreased *Il6*, *Il12b*, and *Il1b* mRNA, and cytosolic pro-IL-1β, and IL-6 and IL-12p40, but not *Tnf* and *Nfkbia* mRNA or TNF protein expression

upon LPS challenge (Supplementary Fig. 1d–h). Nevertheless, LPS-induced IκBα degradation, activation of MAPKs JNK and p38, and nuclear translocation of IRF3 were comparable in control and TUT7-silenced cells (Supplementary Fig. 1i, j). These

**Fig. 1 TUT7 deficiency alters gene expression profile in BMDM response to LPS. a–f** BMDMs from wild-type ($Tut7^{+/+}$) and $Tut7^{-/-}$ mice were incubated with 100 ng/ml LPS for indicated times. TUT7 protein expression was analyzed by immunoblotting (**a**). *Il6, Il12b,* and *Il1b* mRNA expression was determined by RT-qPCR (**b, d**). The levels of indicated inflammatory cytokines in cultural media were analyzed by ELISA (**c, d**). The levels of pro-IL-1β, IκBα, and phosphorylation of MAP kinases were analyzed by immunoblotting (**e, f**). **g** Volcano plot of the changes of innate immune-related gene expressions in $Tut7^{-/-}$ BMDMs after 4 h treatment with 100 ng/ml LPS against wild-type BMDMs. The transcriptome of BMDMs from wild-type and $Tut7^{-/-}$ was compared to baseline. The *x*-axis indicates the logarithm of *p* values to the base 2 of the fold-change (FC) and the *y*-axis reveals the negative logarithm of that to the base 10. Black horizontal dash line indicates significance at 5% criteria (*p* value = 0.05). $\log_2 FC > +0.2$ and $\log_2 FC < -0.2$ indicate increase of transcript levels by >20% and decreased by >20%, respectively. Green and red dots denote transcripts related to innate immune responses that were significantly lower (*n* = 111) and higher (*n* = 68) in response to LPS, respectively. The innate immune-related genes that are significantly differentially expressed are listed in Supplementary Tables 2, 3. **h** 8-week-old mice were injected intraperitoneally with LPS (50 μg/kg). Whole blood was collected 2 h after LPS challenge, and the concentrations of IL-6, IL-12p40, and TNF in sera were determined by ELISA. Data (except **g**) presented are representative of three independent experiments with triplicates in each experiment (error bars, mean ± S.D.). The *p* values were obtained from two-tailed Student's *t* test and are shown in the figure if *p* < 0.05. Source data are provided as a Source Data file.

results suggest a regulatory role for TUT7 in TLR4-mediated immune response.

We generated $Tut7^{-/-}$ mice using the clustered regularly interspaced short palindromic repeats (CRISPR)/Cas9 system (Supplementary Fig. 2 and Supplementary Table 1). Results in Fig. 1b showed that TUT7 deficiency decreased the levels of *Il6, Il12b,* and *Il1b* mRNA and proteins (Fig. 1b, c, e) but not *Tnf* and *Nfkbia* mRNA or TNF protein in BMDMs response to LPS (Fig. 1d). Neither did TUT7 deficiency affect LPS-induced activation of IKK and MAPKs (Fig. 1f).

RNA Sequencing (RNA-Seq) was applied to further assess the effect of TUT7 in global gene expression during TLR4 activation. Surprisingly, TUT7-modulated 352 out of a total 7637 of LPS-induced genes. Among the 352 genes, 111 genes involved in the innate immune response were downregulated in $Tut7^{-/-}$ cells. These include pro-inflammatory cytokines (IL-6, IL-12p40 (*Il12b*), and IL-23p19 (*Il23a*)), cluster of differentiation (CD) markers CD24a (*Cd24a*) and CD55 (*Cd55*), and chemokine CCL17 (*Ccl17*) (Fig. 1g and Supplementary Table 2). By contrast, 68 innate immune-related genes were upregulated in LPS-treated $Tut7^{-/-}$ BMDMs (Fig. 1g and Supplementary Table 3).

$Tut7^{+/+}$ and $Tut7^{-/-}$ mice were challenged with a low-dose LPS. Results in Fig. 1h showed that the serum concentrations of IL-6 and IL-12p40 in $Tut7^{-/-}$ were lower than in $Tut7^{+/+}$ mice after LPS challenge whereas the level of TNF remained comparable (Fig. 1h). These data clearly demonstrate that TUT7 is involved in TLR4-triggered inflammatory responses.

**TUT7 regulation of *Il6* mRNA stability requires its enzymatic activity.** Aspartic acid residues at positions 1058 and 1060 on the TUT7 nucleotidyltransferase domain (Supplementary Fig. 3) are critical for its nucleotidyltransferase activity[19]. To determine whether the nucleotidyltransferase activity of TUT7 is required for modulation of TLR4-driven cytokine production, we reconstituted TUT7-silenced RAW 264.7 macrophages with either human wild-type TUT7 (hTUT7) or a catalytically inactive mutant hTUT7(DADA) whose aspartate residues at positions 1058 and 1060 were replaced by an alanine (Supplementary Fig. 3). While reconstitution with wild-type TUT7 increased LPS-triggered *Il6* and *Il12b*, reconstitution with the hTUT7(DADA) mutant did not and *Tnf* expression was not affected by reconstitution with either wild-type hTUT7 or hTUT7(DADA) (Fig. 2a). Our results showed that nucleotidyltransferase activity is critical for the ability of TUT7 to modulate TLR4-driven cytokine response.

TUT7 and TUT4 are reported to control global mRNA turnover in HeLa cells[20]. Therefore, we hypothesized that TUT7 regulates inflammatory response in macrophages by modulating the stability of cytokine mRNAs. To address this possibility, we investigated *Il6* as an example. The half-life of *Il6* but not that of

*Tnf* in TUT7-silenced macrophages was significantly reduced compared to that in $Tut7^{+/+}$ cells after stimulation with LPS (Fig. 2b). The 3′-UTR plays a critical role in controlling mRNA stability in innate immunity[8,10,34]. To determine whether TUT7 modulates *Il6* turnover via its 3′-UTR, we transfected TUT7-sufficient and -silenced RAW 264.7 cells with luciferase reporter constructs (pGL3) containing either mouse *Il6* (*mIl6*) or human *IL6* (*hIL6*) 3′-UTRs (Fig. 2c). The luciferase activity in TUT7-depleted RAW 264.7 cells in response to LPS was significantly lower when the full-length *mIl6*- or *hIL6*-3′-UTR but not *Tnf*-3′-UTR was transduced compared to that in control cells (Fig. 2d). In the meantime, both control and TUT7-depleted RAW 264.7 cells transduced luciferase reporter driven by *hIL6* promoter variants, AGC and GGG[35], had similar response to LPS (Supplementary Fig. 4). These data together indicate that TUT7 regulates *Il6* expression induced by LPS not at the transcription level but posttranscriptionally via its 3′-UTR.

**TUT7 regulates *Il6* mRNA expression via its 3′-UTR$_{56-104}$.** Several regulatory motifs in the *Il6* 3′-UTR, including five AREs and one stem-loop structure, were reported to control *Il6* mRNA stability[14,36,37]. We transduced luciferase reporter plasmids containing each of the five mutated AREs (ΔARE1-ΔARE5) on *mIl6* 3′-UTR into TUT7-sufficient and -silenced cells. Luciferase activity results showed that mutation in any of the five AREs did not affect TUT7-mediated *Il6* mRNA stability, indicating that AREs are not required for TUT7-modulated TLR4 responsiveness (Supplementary Fig. 5a). We then constructed luciferase reporter plasmids containing different regions of *mIl6* 3′-UTR (Fig. 2c). While expressions of full-length *mIl6* 3′-UTR and *mIl6* 3′-UTR$_{56-173}$ reduced luciferase activities in TUT7-silenced cells compared to TUT7-sufficient cells in response to LPS, *mIl6* 3′-UTR$_{1-70}$ and *mIl6* 3′-UTR$_{172-403}$ did not (Fig. 2e). We further shortened *mIl6* 3′-UTR$_{56-173}$ to *mIl6* 3′-UTR$_{56-113}$, the region containing the conserved stem-loop structure. Similar to full-length *mIl6* 3′-UTR and *mIl6* 3′-UTR$_{56-173}$, transduction of *mIl6* 3′-UTR$_{56-113}$ resulted in lower luciferase activity (Fig. 2f), suggesting that *mIl6* 3′-UTR$_{56-113}$ is possibly the target of TUT7.

Analysis using TargetScan Mouse revealed three putative miRNA (miR-223, miR-224 and miR-376a) sites on *mIl6* 3′-UTR$_{56-104}$ (Supplementary Fig. 5b). We mutated each miRNA target site to study whether TUT7-mediated suppression was dependent on any of the three miRNAs. Our results showed that the suppressive effect of TUT7 was lost when the target site for miR-376a but not miR-223 or miR-224 was mutated (Supplementary Fig. 5c). However, miR-376a target site is only observed in *mIl6* 3′-UTR, but not in *hIL6* 3′-UTR, suggesting that TUT7 control of *Il6* expression is not via miR-376a. Interestingly, the putative miR-376a binding site overlaps with the conserved stem-loop structure. To determine whether the stem-loop structure on

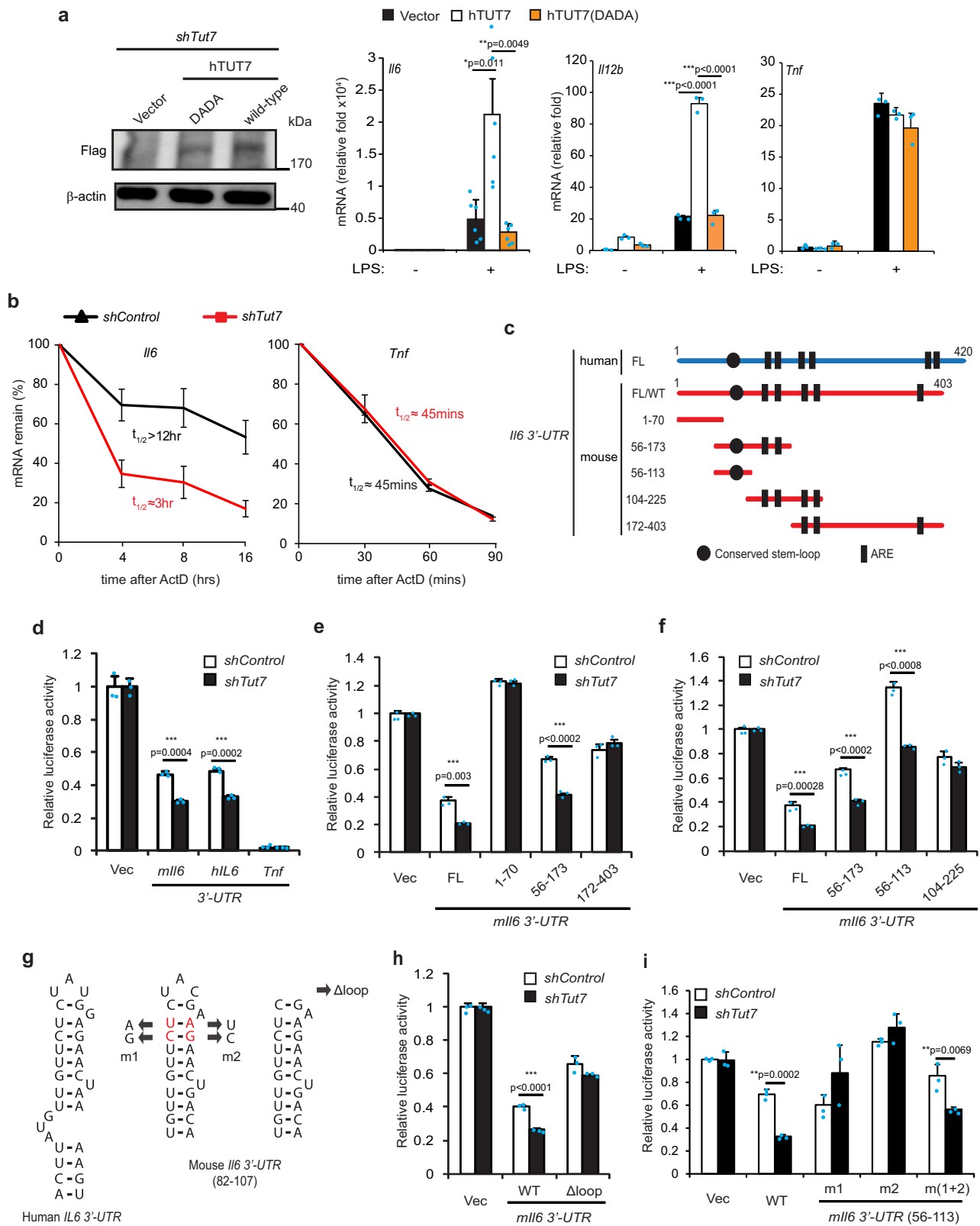

*Il6* is responsible for TUT7-mediated effects, we generated luciferase reporter constructs with either altered or similar stem-loop structures as the wild-type (Fig. 2g). Deletion of the internal loop structure (Δloop) abolished the suppressive effect of

TUT7 (Fig. 2g, h). In addition, m1 and m2 mutants (with altered stem structures) both dampened TUT7-modulated inhibition of the luciferase activity; whereas m(1 + 2) mutant with an intact stem structure that was restored by combining both mutations

**Fig. 2 Regulation of *Il6* mRNA stability by TUT7 is dependent on its 3′-UTR$_{56-104}$. a** *Tut7*-silenced RAW 264.7 macrophages were reconstituted with either flag-tagged human wild-type TUT7 (hTUT7) or activity-dead TUT7 (hTUT7(DADA)) and challenged with 100 ng/ml LPS for 4 h. The expression of IL-6, IL-12p40, and TNF mRNAs and TUT7 protein were analyzed by RT-qPCR and immunoblotting, respectively. **b** RAW 264.7 macrophages expressing *shControl* and *shTut7* were treated with 100 ng/ml LPS for 4 h followed by incubation with 5 μg/ml actinomycin D for the indicated times. Total RNAs were extracted and analyzed by RT-qPCR. **c** Schematic diagram of the *firefly* luciferase reporter constructs containing full-length (FL) and truncated forms of mouse (red line) or human (blue line) *Il6* 3′-UTR. **d–f** Control and *Tut7*-depleted RAW 264.7 cells were co-transfected with the indicated *firefly* luciferase reporter plasmids containing various mouse or human *Il6* 3′-UTR (m*Il6* 3′-*UTR* or h*IL6* 3′-*UTR*) or *Tnf* 3′-UTR and *TK-renilla* control reporter plasmid. At 48 h post-transfection, cells were treated with 100 ng/ml LPS for 8 h and the luciferase activities were determined. **g** Schematic diagram of the predicted stem-loop structure in human and mouse *Il6* 3′-UTR, and mutants whose stem-loop structure (m1 and m2) or loop structure (Δloop) were disrupted or deleted. The stem-loop structure of m*Il6* 3′-*UTR* or h*IL6* 3′-*UTR* was predicted by ref. [14]. Mutant m(1 + 2) was generated to swap sequences on the 5′ and 3′ sides of the stems without disrupting the stem-loop structure in m*Il6* 3′-*UTR*. **h, i** Control and *shTut7*-expressing RAW 264.7 cells were co-transfected with the indicated luciferase reporter plasmids and *TK-renilla* control reporter plasmid for 48 h. Cells were stimulated with 100 ng/ml LPS for 8 h and harvested for the analysis of luciferase activity. Data presented are representative of three independent experiments with triplicates in each experiment (error bars, mean ± S.D.). The *p* values were obtained from two-tailed Student's *t* test and are shown in the figure if $p < 0.05$. Source data are provided as a Source Data file.

---

regained the responsiveness of TUT7 (Fig. 2i). Taken together, TUT7-regulated *Il6* mRNA destabilization is dependent on the stem-loop structure in its 3′-UTR$_{56-104}$.

**TUT7 controls *Il6* mRNA stability through Regnase-1.** Previous studies showed that TUT7 uridylates target RNAs to modulate their stability[20,25,26]. However, our RNA-immunoprecipitation (RNA-IP) data demonstrated that TUT7 did not associate with *Il6* transcripts (Fig. 3a). Regnase-1 is reported to bind and control *Il6* stability via the conserved stem-loop structure in *Il6* 3′-UTR$_{56-104}$[37]. Therefore, we hypothesized that TUT7 controls *Il6* mRNA destabilization through regulating Regnase-1 expression. Results showed that the levels of mRNA and protein of Regnase-1 increased in LPS-stimulated TUT7-depleted RAW 264.7 cells (Fig. 3b, c) and BMDMs (Fig. 3d, e) compared with that in control cells. Regnase-1 belongs to the Zc3h12 family that is characterized by its conserved CCCH-type zinc-finger[15]. The mRNA level of another family member *Zc3h12c* was not altered in LPS-treated cells, regardless of the presence or absence of TUT7 (Supplementary Fig. 6), suggesting a specific regulatory role of TUT7 in *Zc3h12a* expression. The half-life of *Zc3h12a* was longer (30 vs. 80 min) in TUT7-deficient RAW 264.7 macrophages than control cells upon LPS challenge (Fig. 3f). In addition, LPS-induced *Zc3h12a* expression was reduced in TUT7-depleted cells reconstituted with wild-type TUT7, but not with hTUT7(DADA) mutant (Fig. 3g), indicating that TUT7-mediated *Zc3h12a* expression after LPS stimulation depends on its nucleotidyltransferase activity. To further confirm TUT7-mediated TLR4-induced cytokine production is through Regnase-1, we depleted Regnase-1 in *Tut7*$^{-/-}$ BMDMs by lentivirus-mediated shRNA and challenged these cells with LPS. Deletion of Regnase-1 in *Tut7*$^{-/-}$ BMDMs significantly increased *Il6* and *Il12b* expression in response to LPS (Fig. 3h). However, LPS-induced *Tnf* and *Nfkbia* were similar in *Tut7*$^{-/-}$ BMDMs expressing control and *Zc3h12a* shRNA (Fig. 3h). These data together demonstrate that TUT7 regulates TLR4-induced *Il6* and *Il12b* expression by suppressing Regnase-1 expression.

**TUT7 controls *Zc3h12a* mRNA stability through its 3′-UTR.** We then determined whether TUT7 controls *Zc3h12a* mRNA stability via its 3′-UTR using luciferase reporter assay. We generated a set of luciferase reporter constructs containing full-length or different truncation of the *Zc3h12a* 3′-UTR and analyzed their luciferase activities in control and TUT7-depleted RAW 264.7 cells (Fig. 4a). The luciferase activity of full-length *Zc3h12a* 3′-UTR was upregulated in LPS-stimulated TUT7-depleted RAW 264.7 cells compared to that in control cells (Supplementary

Fig. 7). To further assess whether TUT7 directly regulates *Zc3h12a* expression via its 3′-UTR, luciferase reporter constructs together with either wild-type or enzymatic inactive TUT7 were co-expressed in human embryonic kidney 293 (HEK 293) cells. Like most mRNA 3′-UTRs, full-length mouse *Zc3h12a* 3′-UTR decreased luciferase activity compared to luciferase reporter control (Fig. 4b). This repressive effect was further enhanced by co-expression of wild-type TUT7, but not the hTUT7(DADA) mutant, suggesting that TUT7 directly regulates *Zc3h12a* in a nucleotidyltransferase activity-dependent manner (Fig. 4b). In addition, while TUT7 overexpression reduced *Zc3h12a* 3′-UTR$_{1-358}$ luciferase activity, it failed to reduce the luciferase activities of *Zc3h12a* 3′-UTR$_{338-698}$ and *Zc3h12a* 3′-UTR$_{560-865}$ (Fig. 4b). We further shortened *Zc3h12a* 3′-UTR$_{1-358}$ and found that *Zc3h12a* 3′-UTR$_{100-250}$ rendered TUT7-mediated suppressive effect similar to full-length *Zc3h12a* 3′-UTR and *Zc3h12a* 3′-UTR$_{1-358}$ (Fig. 4c). *Zc3h12a* 3′-UTR$_{100-250}$ was previously predicted to be able to form a stem-loop structure[11]. We then investigated the role of this stem-loop structure in TUT7-modulated *Zc3h12a* mRNA stability by constructing luciferase reporters containing *Zc3h12a* 3′-UTR$_{100-250}$ mutants whose stem-loop structures were eliminated (Fig. 4d)[11]. Surprisingly, disruption of the stem-loop structure (St1m and St2m) lost TUT7-mediated inhibition of the luciferase activity, whereas restoring stem-loop structure (St(1 + 2)m) regained the effect of TUT7 (Fig. 4e), suggesting a requirement for the stem-loop structure of *Zc3h12a* 3′-UTR in TUT7-mediated suppression. Taken together, TUT7 modulates *Zc3h12a* mRNA stability by recognition of the stem-loop structure in its 3′-UTR$_{100-250}$.

**TUT7 binds and uridylates *Zc3h12a*.** Our results in Fig. 3g revealed that TUT7 directly modulated *Zc3h12a* mRNA stability through its 3′-UTR in an uridyltransferase activity-dependent manner. TUT7 possesses the ability to uridylate both mRNA and miRNA and to facilitate their degradation[20,23]. We thus hypothesized that TUT7 uridylates and degrades *Zc3h12a* after TLR4 engagement. To test this possibility, we first did RNA-IP to assess the association of TUT7 and *Zc3h12a*. Results in Fig. 5a, b showed that TUT7 was associated with *Zc3h12a* by 2 h after LPS treatment.

To further investigate whether TUT7 directly binds to *Zc3h12a* stem-loop structure, we performed RNA electrophoresis mobility shift assay (EMSA) using synthesized RNA probes. In line with the results of RNA-IP, TUT7 bound to murine *Zc3h12a* stem-loop, but not that of *Il6* (Fig. 5c). A previous study showed that the length of poly(A) tails negatively correlates with the frequency of uridylation[20]. Interestingly, TUT7 bound to *Zc3h12a* stem-loop regardless of the length of poly(A) tails (Supplementary

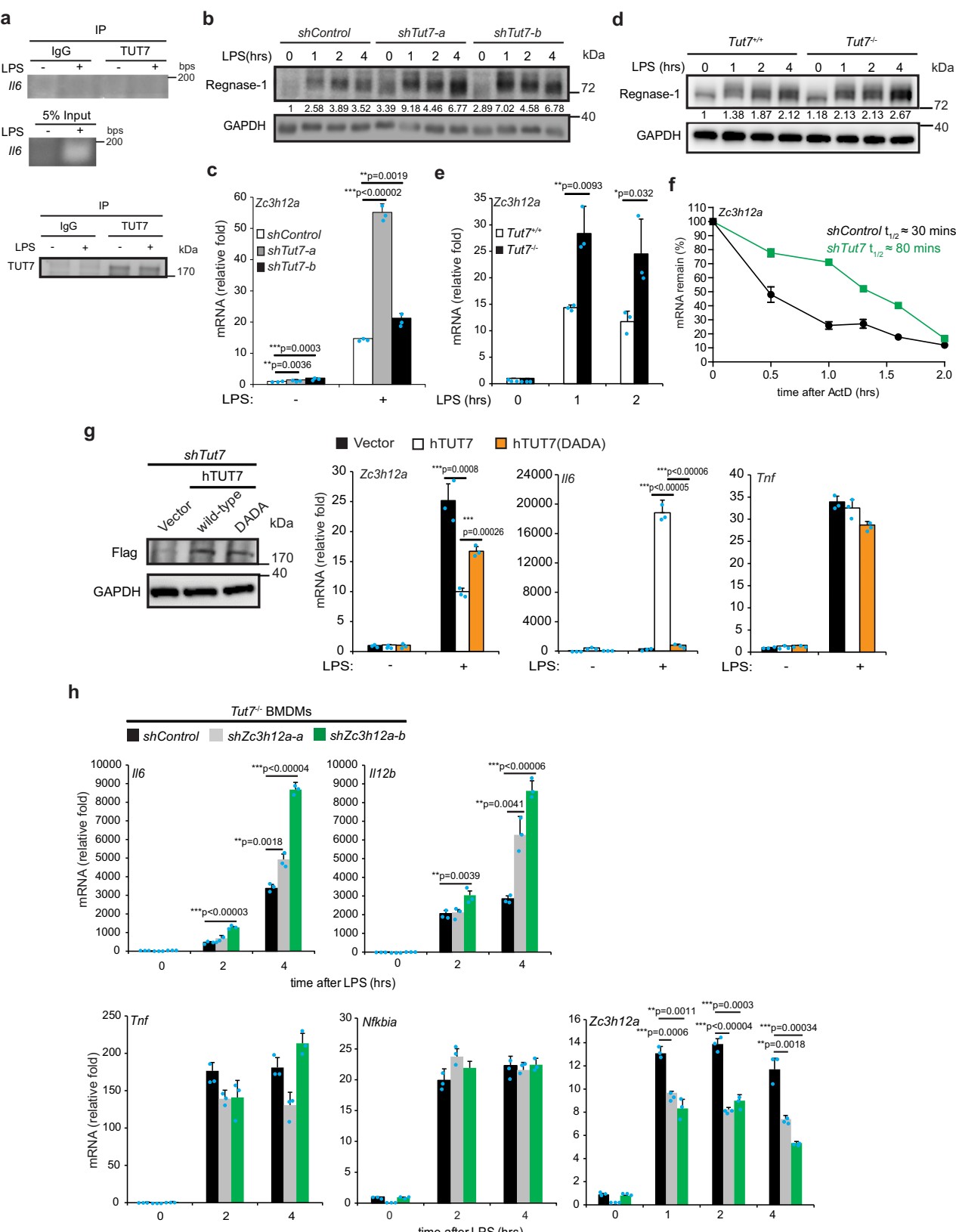

Fig. 8c), suggesting that TUT7 associating with *Zc3h12a* is dependent on its stem-loop structure rather than the length of its poly(A) tails. Furthermore, the association of TUT7 and *Zc3h12a* was dependent on its stem-loop structure rather than its sequence as the addition of RNA oligomers containing wild-type *Zc3h12a* or St(1 + 2)m, but not St1m nor St2m, stem-loop mutant abolished their binding (Fig. 5d). To assess whether TUT7 is able to uridylate *Zc3h12a*, we set up in vitro uridylation assay

**Fig. 3 TUT7 decreases *Zc3h12a* mRNA stability in response to LPS challenge. a** RAW 264.7 cells were treated with 100 ng/ml LPS for 2 h and cell lysates were incubated with control IgG or protein A-agarose beads-conjugated anti-TUT7 antibody at 4 °C for 4 h. TUT7-interacting RNAs were extracted followed by RT-PCR (upper panel). Immunoprecipitation of TUT7 was examined by immunoblotting (lower panel). **b–e** Control (*shControl*) and *Tut7*-knockdown RAW 264.7 cells and wild-type and *Tut7⁻/⁻* BMDMs were treated with 100 ng/ml LPS for different periods of time. The expression of Regnase-1 protein (**b**, **d**) and *Zc3h12a* mRNA (**c**, **e**) were analyzed by immunoblotting and RT-qPCR, respectively. **f** Control- and *shTut7*-expressing RAW 264.7 cells were treated with 100 ng/ml LPS for 2 h followed by treatment with 5 μg/ml of actinomycin D for the indicated periods of time. Regnase-1 mRNA expression was analyzed by RT-qPCR. **g** *Tut7*-knockdown RAW 264.7 cells were reconstituted with either hTUT7 or hTUT7(DADA) and challenged with 100 ng/ml of LPS for 2 h. Cells were collected, and the expression of the indicated proteins and mRNAs were analyzed by immunoblotting and RT-qPCR, respectively. **h** Control- and *shZc3h12a*-expressing *Tut7⁻/⁻* BMDMs were stimulated with 100 ng/ml LPS for the indicated periods of time. Total RNAs were prepared and the mRNA expression of the indicated genes was measured by RT-qPCR. Data presented are representative of three independent experiments with triplicates in each experiment (error bars, mean ± S.D.). The *p* values were obtained from two-tailed Student's *t* test and are shown in the figure if *p* < 0.05. Source data are provided as a Source Data file.

using immunopurified TUT7 from HEK293T cells overexpressing Flag-tagged TUT7. RNA substrates were in vitro transcribed *Zc3h12a* or *Tnf* containing 3′-UTR (Fig. 5e). Interestingly, regardless of the coding sequences (CDSs) of the substrates, TUT7 was only capable of uridylating transcripts containing *Zc3h12a* 3′-UTR, but not those without 3′-UTR or those with *Tnf* 3′-UTR (Fig. 5f). Consistently, TUT7 did not uridylate transcripts containing *Il6* 3′-UTR albeit with the CDS of *Zc3h12a* (Fig. 5g). Furthermore, TUT7 preferentially uridylated transcripts containing *Zc3h12a* stem-loop rather than that containing *Il6* stem-loop (Fig. 5h). Pull-down and immunoblotting experiments showed that Regnase-1 was not co-immunoprecipitated with TUT7 (Supplementary Fig. 8a, b), which excludes the possibility of the involvement of Regnase-1 in TUT-mediated uridylation. These data together support the notion that upon LPS stimulation TUT7 directly binds and uridylates *Zc3h12a* 3′-UTR.

We further examined *Zc3h12a* 3′-terminal uridylation in *Tut7⁺/⁺* and *Tut7⁻/⁻* BMDMs using gene specific TAIL-Seq[31] (Fig. 6a). As expected, LPS significantly increased the frequency of uridylation on *Zc3h12a*, especially that of oligo-uridylation (≥2 U) (Fig. 6b). Deleting TUT7 significantly decreased uridylation on *Zc3h12a* (12% mono-uridine and 14% oligo-uridines among 1714 reads) after LPS stimulation compared to that of control (14% mono-uridine and 24% oligo-uridines among 1067 reads) (Fig. 6b). However, there is no observable difference in uridylation on *Il6* mRNA between *Tut7*-deficient macrophages (9% mono-uridine and 5% oligo-uridine among 1730 reads) and control (9% mono-uridine and 6% oligo-uridine among 1756 reads) in response to LPS (Supplementary Fig. 9). Consistent with published studies[20,31], the frequency of oligo-uridine (≥2 U) addition to *Zc3h12a* 3′ end negatively correlated with the length of poly(A) tails in wild-type cells but not in TUT7-deficient cells upon LPS challenge (Fig. 6c). Furthermore, TUT7 deficiency did not affect non-templated addition of cytosine and guanine to the 3′ end of *Zc3h12a* (Fig. 6d). Together, our results demonstrate that TUT7 binds *Zc3h12a* in macrophage stimulated by LPS and regulates *Zc3h12a* expression by uridylating its 3′ tail.

**TUT4 and TUT7 regulate *Zc3h12a* expression through different mechanisms.** TUT7 and TUT4 are highly homologous among members of the TUTase family. They have been reported to share various similar biological functions[38]. We thus examined whether TUT4 like TUT7 possesses the modulatory function of TLR4-driven effect through regulating *Zc3h12a* expression. We first assessed TUT4 expression in *Tut7*-deficient cells and found that the levels of TUT4 mRNA and protein were similar in TUT7-silenced and control cells (Supplementary Fig. 10a, b). However, unlike TUT7, depletion of TUT4 only marginally reduced Regnase-1 mRNA and protein expression after LPS stimulation (Supplementary Fig. 10c, d), indicating that TUT4 and

TUT7 participates in TLR4-triggered inflammatory responses through different mechanisms.

## Discussion

Inflammatory cytokines are quickly induced upon infection in order to eliminate pathogens and repair damaged tissues. To prevent their detrimental effect on the host, their production must be tightly regulated. PTR is a quick and effective mechanism to modulate the expression of inflammatory cytokines[6]. The involvement of PTR in regulation of cytokine production during innate immune response is recently emerged. As shown in Fig. 7, we reveal that terminal uridyltransferase TUT7 expression is induced by TLR4 ligand LPS, and, it in turn participates in the regulation of the expression of a subset of cytokines, including IL-6 and IL-12β. TUT7 positively regulates *Il6* mRNA stability through controlling *Zc3h12a* expression by its terminal nucleotidyltransferase activity after TLR4 activation. We further demonstrate that upon TLR4 engagement TUT7 directly binds and uridylates *Zc3h12a* to downregulate its expression, resulting in stabilization of *Il6* mRNAs. Our results together demonstrate that TUT7 is a modulator of TLR4-triggered inflammatory response (Fig. 7). TUT7 fine-tunes the expression of IL-6, and we speculate other inflammatory cytokines as well, through directly uridylating *Zc3h12a* transcript.

Accumulating data have revealed that a number of TLRs-inducible proteins play a role in modulation of inflammatory responses[39]. In this study, we extend this list to include TUT7. Our data indicate that TLR4-induced TUT7 expression requires IKK activity, and p38, but not JNK nor ERK, MAPK also contributes to its expression. These results suggest that transcription factors downstream of IKK and p38 MAPK may be required for TLR4-induced TUT7 expression. We and others have previously reported that p38 MAPK modulates the expression of certain LPS-induced genes via several transcription factors, including CCAAT/Enhancer binding protein β (C/EBPβ) and cAMP response element-binding protein (CREB)[40,41]. Interestingly, online prediction tool PROMO (http://alggen.lsi.upc.es/cgi-bin/promo_v3/promo/promoinit.cgi?dirDB=TF_8.3) identified the conserved binding motifs for C/EBPβ and NF-κB on both human and mouse TUT7 promoters although it remains to be determined whether p38 MAPK regulates TLR4-triggered TUT7 expression through C/EBPβ.

Regnase-1 is an RNase. By controlling the degradation of target mRNAs, such as *Il6* and *Il12b*, Regnase-1 is crucial for restraining inflammatory response during TLRs and IL-1 receptor (IL-1R) activation[37]. Therefore, the dynamics of Regnase-1 expression following TLRs and IL-1β stimulation ensures timely regulation of cytokine production. Accumulating data revealed that both PTR and PTM contribute to the regulation of Regnase-1 expression upon TLRs- and IL-1R activation[37]. Regnase-1 recognizes the conserved stem-loop structure in its 3′-UTR to

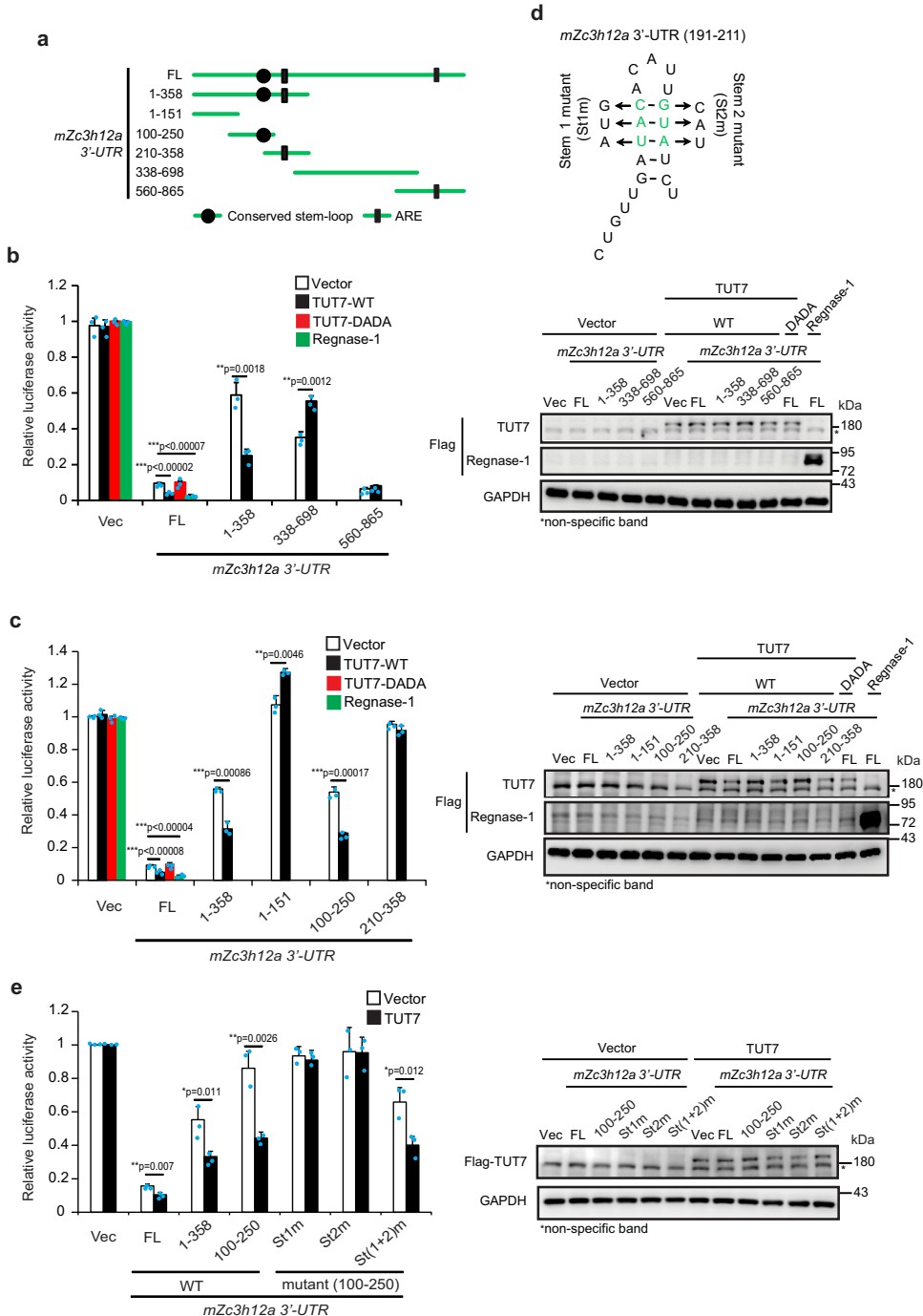

**Fig. 4 TUT7 modulates *Zc3h12a* mRNA stability via its 3′-UTR$_{100-250}$. a** Schematic diagram of the *firefly* luciferase reporter constructs containing full-length (FL) and truncated forms of mouse *Zc3h12a* 3′-UTR (*mZc3h12a* 3′-UTR). **b, c, e** HEK 293 cells were co-transfected with wild-type TUT7 (TUT7-WT), enzymatic inactive TUT7 (TUT7-DADA), or Regnase-1 expression plasmids, the indicated *firefly* luciferase reporter constructs containing various fragments of *mZc3h12a* 3′-UTR as well as *TK-renilla* control reporter plasmid. At 48 h post-transfection, cells were harvested and the luciferase activities were determined. The expression of indicated proteins was confirmed by immunoblotting and is shown in the right panel. **d** Schematic diagram of the *firefly* luciferase reporter constructs containing stem mutants (St1m and St2m) of *mZc3h12a* 3′-UTR$_{100-250}$. Mutant St(1 + 2)m was generated by swapping sequences on both stems in *mZc3h12a* 3′-UTR. Mutations of the stem-loop structure of *mZc3h12a* 3′-UTR$_{100-250}$ were predicted by ref. [11]. Data presented are representative of three independent experiments with triplicates in each experiment (error bars, mean ± S.D.). The *p* values were obtained from two-tailed Student's *t* test and are shown in the figure if *p* < 0.05. Source data are provided as a Source Data file.

degrade its own mRNA at the early phase of an inflammatory response[11]. Interestingly, Regnase-1 protein is quickly phosphorylated and degraded through IKK complex upon TLRs and IL-1R activation[11]. It may thereby unable to effectively eliminate its own mRNA, impeding the subsequent cytokine production.

Our findings demonstrate that TUT7 induces *Zc3h12a* degradation through uridylation on its 3′ end in the early phase of TLR4-triggered inflammatory response. Our data together with a previous study[11] indicate that at least two distinct PTR mechanisms triggered by TLR4 destabilize *Zc3h12a* to promote the production

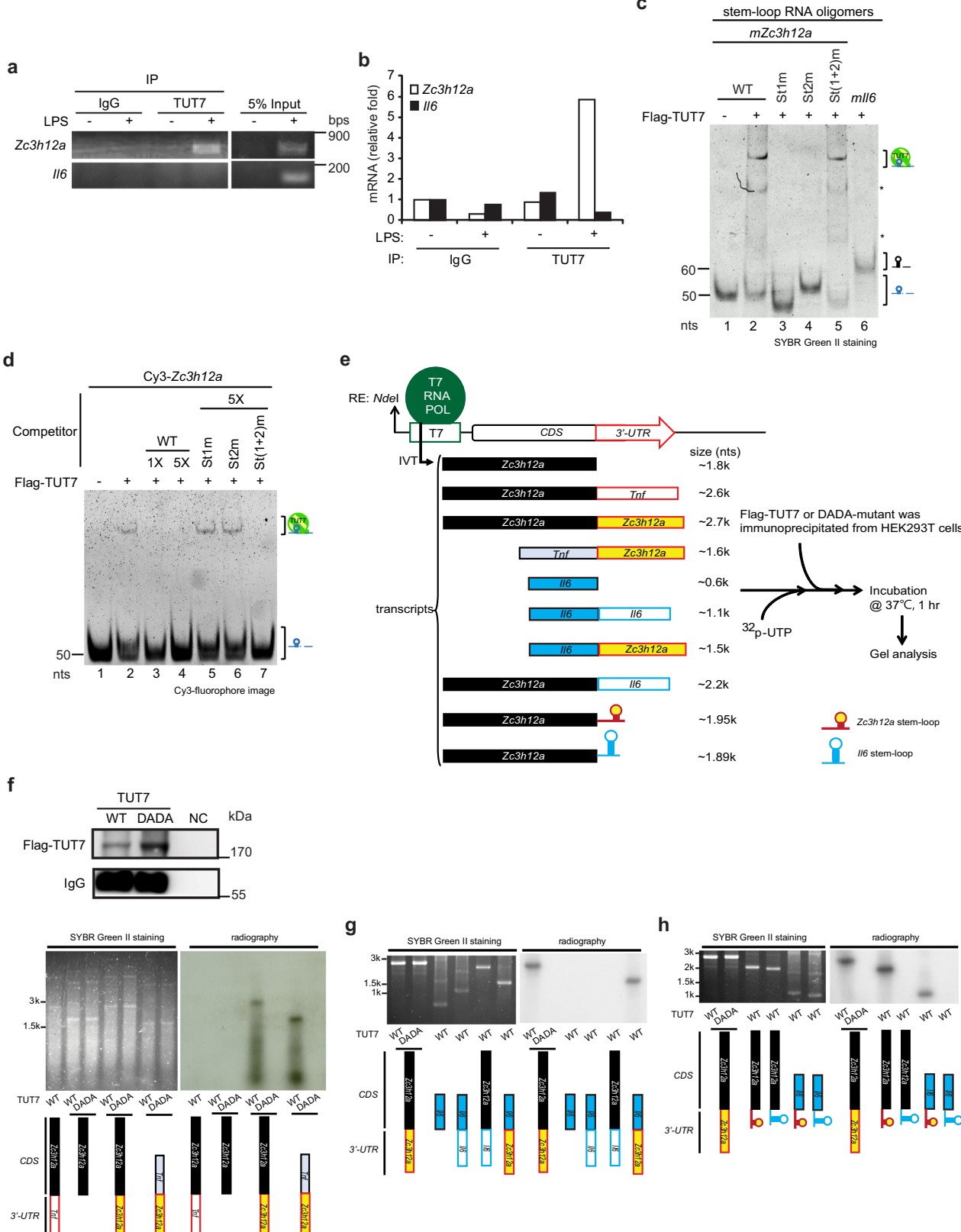

of cytokines and inflammatory mediators in the early phase of innate immune responses.

Among the members of the TUTase family, TUT7 shares structural and functional homology with TUT4. Both TUT7 and

TUT4 function to uridylate noncoding RNAs, i.e., miRNAs, leading to biogenesis or degradation of miRNA[22,23,27,42]. Recent reports indicate that TUT7 and TUT4 act redundantly in NIH3T3 and HeLa cells to control the stability of global mRNAs

**Fig. 5 TUT7 binds and uridylates *Zc3h12a*. a, b** RAW 264.7 cells were treated with 100 ng/ml LPS for 2 h and cell lysates were incubated with control IgG or protein A-agarose beads-conjugated anti-TUT7 antibody at 4 °C for 4 h. TUT7-interacting RNAs were extracted and analyzed by RT-PCR (**a**) and RT-qPCR (**b**). **c** Synthetic RNAs containing *Il6* or *Zc3h12a* wild-type stem-loop or mutant variants were incubated with immunopurified Flag-tagged wild-type TUT7 at 37 °C for 30 min. **d** Cy3-labeled RNA containing *Zc3h12a* stem-loop was incubated with immunopurified TUT7 in the presence or absence of 1× or 5× fold excess of unlabeled RNA containing *Zc3h12a* wild-type or the indicated stem-loop mutant at 37 °C for 30 min. All RNA samples were resolved on a 10% native PAGE and visualized by SYBR Green II staining (**c**) and Cy3 fluorescent signal (**d**). **e** Schematic diagram of different constructs used for in vitro transcription. Flowchart on the right shows the procedure for in vitro uridylation. RNA transcripts were prepared by in vitro transcription. Transcripts were incubated with Flag-tagged wild-type (WT) or enzyme-dead (DADA) TUT7 mutant prepared by immunoprecipitation of HEK293T cells transfected with Flag-tagged TUT7- or DADA mutant-expressing plasmids in the presence of α-$^{32}$p-UTP at 37 °C for 1 h. **f–h** Immunoprecipitation of flag-tagged wild-type and DADA mutant TUT7 was confirmed by immunoblotting (the upper panel of **f**). RNA samples were separated on a 1% formaldehyde-agarose gel and visualized by SYBR Green II staining and autoradiography (the bottom panel of **f**, **g**, **h**). Data presented are representative of three independent experiments. Source data are provided as a Source Data file. Asterisk (*) represents the nonspecific binding.

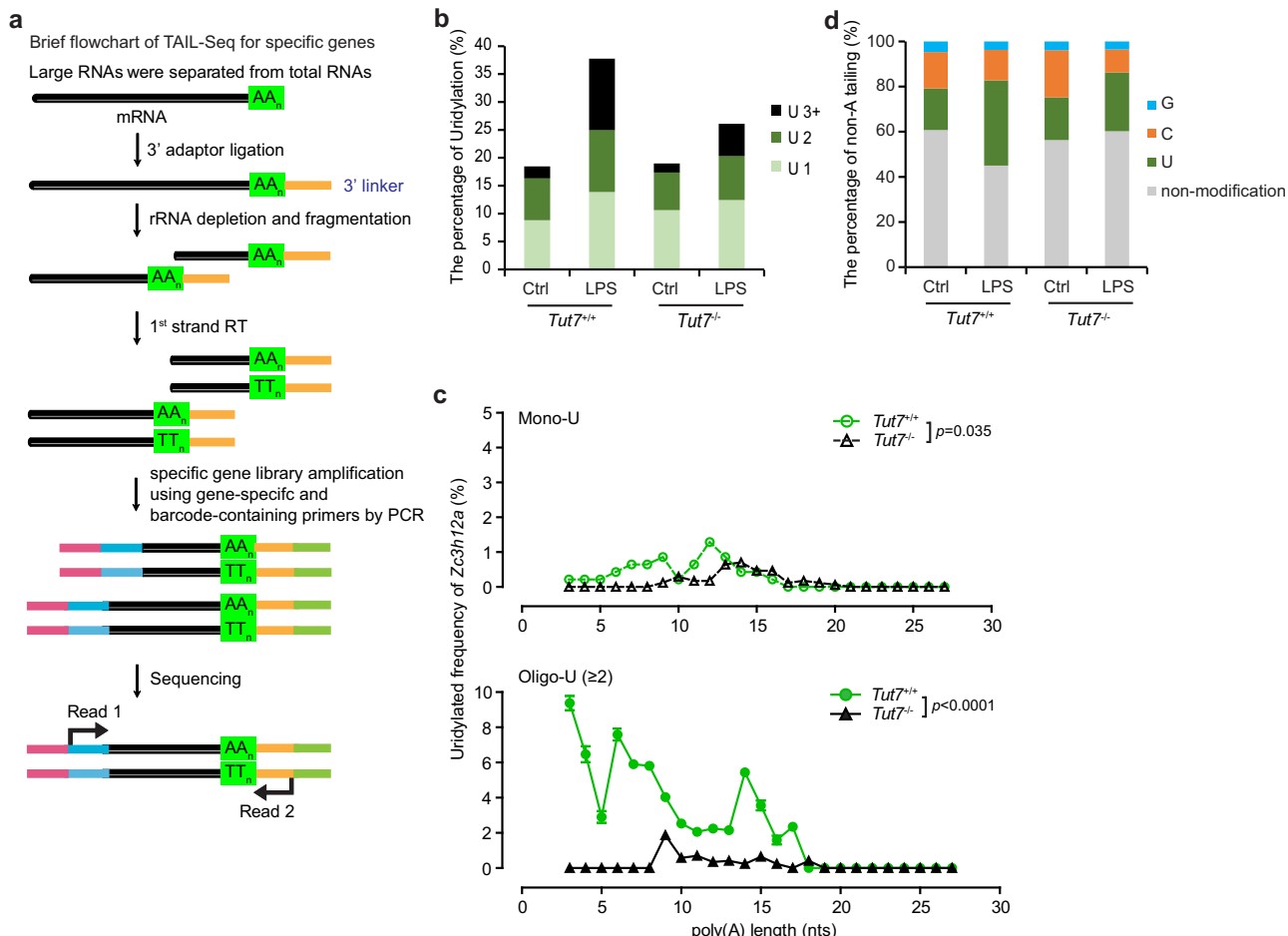

**Fig. 6 TUT7 mediates oligo-uridylation of *Zc3h12a* on 3′ end in response to LPS. a** Flowchart showing the experimental procedure of specific gene TAIL-Seq to analyze 3′ tails of *Zc3h12a* mRNA transcripts. **b** The frequency of *Zc3h12a* uridylation assessed by specific gene TAIL-Seq. Uridylation frequency is the percentage of uridylated *Zc3h12a* reads out of the total number of *Zc3h12a* reads. **c** the correlation between the mono-uridylation (upper panel) and oligo-uridylation (lower panel) of *Zc3h12a* and their poly(A) tail lengths. Uridylation frequency presents the percentage of uridylated reads among the total *Zc3h12a* reads. **d** The frequency of non-templated 3′ end modifications on *Zc3h12a* tails. The *Y*-axis presents the percentage of uridine, cytosine, and guanine additions among total *Zc3h12a* reads (for cytosine and guanine additions, 75 and 22 among 466 reads, 144 and 40 among 1067 reads, 65 and 12 among 311 reads, and 174 and 60 among 1714 reads in *Tut7*$^{+/+}$, *Tut7*$^{+/+}$(+LPS), *Tut7*$^{-/-}$, and *Tut7*$^{-/-}$(+LPS) cells, respectively. The value of uridylation is from **b**). The *p* value in (**c**) was assessed by Pearson correlation analysis. Source data are provided as a Source Data file.

through uridylation, especially oligo-uridylation (≥2 U) as oligo-uridylation is crucial for mRNA decay[20,31]. Both TUTs also function redundantly to eliminate both maternal mRNA during early embryogenesis and viral RNAs in mammalian cells[28,43,44]. Our results show that upon TLR4 activation, TUT7, but not TUT4, is upregulated and TUT7 subsequently associates with and oligouridylates *Zc3h12a* to regulate cytokine production. On contrast, TUT4 positively regulates *Zc3h12a* expression response to LPS through a yet-to-be investigated mechanism. TUT7 and TUT4 were reported to regulate *Il6* expression through uridylation of miR-26, yet miR-26 uridylation is pervasive[24,32]. Our study demonstrates that mRNA uridylation can be specific and

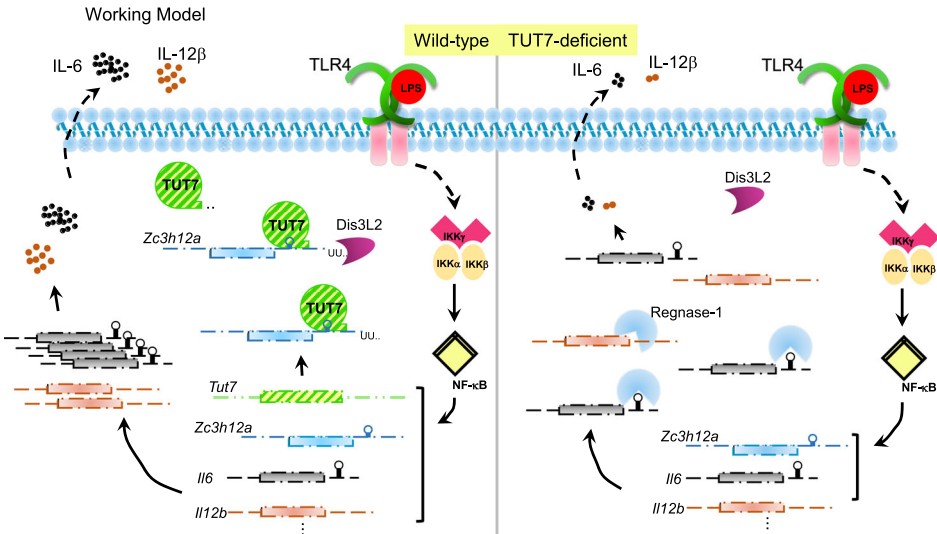

**Fig. 7 Model for the regulation of TUT7-mediated Regnase-1 expression.** In response to TLR4 activation, IKK complex phosphorylates IκBα to promote NF-κB-driven transcription. Messenger RNAs of IL-6, IL-12β, Regnase-1, and TUT7 are upregulated. TUT7 targets and uridylates *Zc3h12a* on the stem-loop structure of *Zc3h12a* 3′-UTR. Uridylated *Zc3h12a* transcripts are eliminated by exonuclease, probably Dis3L2, leading to upregulation of the expression of a subset of cytokines, including IL-6 and IL-12β, at the early stage during TLR4-triggered inflammation. When TUT7 is depleted, elevated Regnase-1 results in a substantial decrease in IL-6 and IL-12β expression. The dashed and solid lines indicate indirect and direct pathways, respectively.

inducible and that TUT7 is functionally distinct from TUT4 in innate immune responses. However, it is worth noting that LPS-induced *Zc3h12a* oligo-U-tails is not completely diminished in *Tut7*-deficient cells, indicating the involvement of additional TUT in *Zc3h12a* uridylation response to LPS.

It is still unclear how TUT4 and TUT7 discriminate and mediate uridylation on the 3′ end of their target mRNAs. Generally, it is proposed that when the poly(A) tails of the target mRNAs become shorter than 20 nts, poly(A)-binding proteins (PABPs) are dissociated from the poly(A) tails, thereby allowing the binding of TUTs to mRNAs for subsequent uridylation[31,45]. This model is based on the assumption that uridylation by TUTs is an activity downstream of deadenylation but it excludes the possibility that TUTs may participate in deadenylation. In addition, it remains to be determined whether any *cis*-element on 3′-UTR is required for TUTs-mediated uridylation. Our data reveal that TUT7 directly binds to *Zc3h12a* through the conserved stem-loop structure in its 3′-UTR and catalyzes uridylation, which probably marks the mRNA for degradation.

Regnase-1 is also known to target the same conserved stem-loop structure as TUT7 does to facilitate its own mRNA decay[11,16]. It is unclear why Regnase-1 can recognize the stem-loop structures appearing in both *Zc3h12a* and *Il6* 3′-UTRs, whereas TUT7 only interacts with *Zc3h12a*, but not *Il6*, stem-loop structure. Structural and functional studies suggest that Regnase-1 prefers binding to the stem-loop structure with a UAU loop and 5 to 9 base pairs in the stems[46]. Interestingly, the stem-loop structures in *Il6* and *Zc3h12a* 3′-UTR are distinct: *Il6* stem-loop structure harbors a typical Y-R-Y loop and 9 base pairs in the stem, whereas the one in *Zc3h12a* 3′-UTR is composed of a 5-nucleotide loop and 5 base pairs in the stem[16]. A recent study employing Hamiltonian scores to predict how TUT7 distinguishes its target pre-let-7a from other unrelated RNAs suggests a critical role of the loop, but not the stem, structure for TUT7 binding[47]. These data together suggest that TUT7 may prefer to interact with the loop in *Zc3h12a* rather than *Il6* 3′-UTR, but this notion needs confirmation. In addition, Lin28 is shown to recognize a GGAG loop and subsequently recruit TUT4 to pre-

let-7[48]. UPF1 helicase is also reported to cooperate with Regnase-1 to control *Zc3h12a* destabilization[16,49]. Therefore, we cannot exclude the possibility that an unknown RNA-binding molecule associates with the stem-loop structure of *Zc3h12a* and enlists TUT7.

Oligo-uridylation on mRNAs in mammalian cells is considered to be a tag for RNA degradation[20,29]. Addition of uridine to the 3′ end of RNAs by TUT7 and TUT4 induces 5′-to-3′ and 3′-to-5′ degradation by the XRN family of exoribonucleases and the exosome complex, respectively[20,23]. The 3′-to-5′ exoribonuclease Dis3L2 has been shown to form complex with TUT4/7 to degrade cytosolic noncoding RNAs, independent of the exosome[50]. However, recent studies indicate that Dis3L2 only plays a modest role in human cells to degrade global mRNAs with uridylated tails[51]. Rather it is a predominant exoribonuclease to eliminate uridylated mRNA during apoptosis[51].

Here we reveal that LPS-induced TUT7 uridylates *Zc3h12a* and accelerates its mRNA degradation thereby promotes the production of its target cytokines. It appears that this regulatory mechanism to suppress Regnase-1 expression is at work in the early phase of inflammation when mRNAs of cytokines, such as IL-6 and IL-12β, are being actively translated. Inflammation is a highly dynamic and intricate process. Multiple factors collaborate to spatially and temporally control the production of inflammatory mediators at the levels of transcription, PTR, or PTM. A recent study demonstrated that Regnase-1 functions to degrade translationally active cytokine mRNAs during inflammation[16]. In addition, our transcriptome analysis revealed that 111 and 68 innate-related genes were up- and down-regulated, respectively, in *Tut7*[−/−] cells upon TLR4 activation. TUT7 therefore may act in concert with other molecules to fine tune the expression of different sets of cytokine mRNAs. How different molecules and mechanisms work together to elaborately control inflammatory response is worth further investigation.

In summary, our study uncovers TUT7-mediated *Zc3h12a* uridylation as a posttranscriptional mechanism in regulation of TLR4-driven inflammatory cytokine response. This molecular mechanism involving TUT7-restricted Regnase-1 expression may

contribute to the generation of a specific set of inflammatory cytokines to shape a beneficial inflammation. Given the prominent role of IL-6 in inflammation, the TUT7/Regnase-1 axis pathway may also contribute to the pathogenesis of many inflammatory diseases.

## Methods

**Plasmids**. The coding regions (CDS) of mouse TUT7, TNF, and Regnase-1 were amplified by PCR using the cDNA library prepared from LPS-treated RAW 264.7 macrophages as templates and cloned into pcDNA3 (Invitrogen, Carlsbad, CA) containing an N-terminal Flag tag. The 3′-UTR of mouse *Zc3h12a*, *Il6* and *Tnf* were also PCR amplified from RAW 264.7 cDNA library and subcloned to the immediate downstream of the CDS of TNF, IL-6 or Regnase-1 in pcDNA to obtain pcDNA-Zc3h12aCDS, pcDNA-Zc3h12aCDS + 3′-UTR, pcDNA-Zc3h12aCDS_TNF3′-UTR, pcDNA-TNFCDS_Zc3h12a3′-UTR, pcDNA-Zc3h12aCDS_IL63′UTR, pcDNA-IL6CDS, pcDNA-IL6CDS + 3′-UTR, IL6CDS_Zc3h12a3′-UTR, pcDNA-Zc3h12aCDS_IL63′UTR-stem loop, and pcDNA-Zc3h12aCDS + 3′UTR-stem-loop plasmids used for in vitro transcription. Human TUT7 and TUT4 cDNAs were amplified by PCR using pCK-FLAG-TUT7 and pCK-FLAG-TUT4 plasmids[20] kindly provided by Dr. V. Narry Kim (National University of Seoul, Korea) as templates and cloned into the pLKO_AS2.neo vector (National RNAi Core Facility, Academia Sinica, Taiwan). Luciferase reporter plasmids pGL3 and pRL-TK were obtained from Promega (Madison, WI). The pGL3-IL-6 3′-UTR full-length (1–403), truncated constructs (1–70, 56–173, and 172–403), and AU-rich elements mutants (ΔARE1-5)[36] were kindly provided by Dr. Keith L. Kirkwood (University of Michigan, MI). Human IL-6 3′-UTR full-length fragment was obtained by restriction digestion of pMIR-human IL-6 3′-UTR plasmid[24] (a kind gift from Dr. Joseph P. Mizgerd, Boston University School of Medicine, MA) using HindIII and SpeI followed by blunting with Klenow, and then cloned into pGL3 plasmid. The pGL3-Regnase-1 3′-UTR full-length plasmid[11] was gifted from Dr. Shizuo Akira (Osaka University, Japan). The pGL3-IL-6 3′-UTR truncated mutants (56–113 and 104–225) and pGL3-Regnase-1 3′-UTR deletion plasmids (1–358, 1–151, 100–250, 210–358, 338–698, and 560–865) were generated by PCR amplified using the pGL3-IL-6 3′-UTR and pGL3-Regnase-1 3′-UTR full length as templates and cloned into pGL3-control vector. The enzyme-dead TUT7, hTUT7-DADA mutant, whose two aspartate residues at 1058 and 1060 were mutated to alanine, various mouse IL-6 3′-UTR$_{56-104}$ and Regnase-1 3′-UTR$_{1-358}$ mutant constructs containing mutations in the stem-loop and putative miRNAs target sites were generated by site-directed mutagenesis using the QuickChange II Site-Directed Mutagenesis Kit (Stratagene, La Jolla, CA). Mutations of the stem-loop structure of *mIl6* 3′-UTR and *mZc3h12a* 3′-UTR$_{100-250}$ were generated by referring the studies[11,14,37]. All mutations were confirmed by DNA sequencing. All primers used in plasmid construction were listed in Supplementary Table 4.

**Reagents and antibodies**. Anti-TUT7 polyclonal antibody (1:1000 for immunoblotting and 1:50 immunoprecipitation) was produced by immunizing rabbits with a recombinant mouse TUT7 peptide (amino acid residues 2001-2828) which was expressed in *Escherichia coli* BL21 and subsequently affinity-purified using the nickel-coated agarose beads, Ni-NTA (Qiagen, Hilden, Germany), following the manufacturer's protocol. Rabbit antisera were collected and further purified by incubation with polyvinylidene fluoride (PVDF) membrane strips (Merck Millipore, Billerica, MA) to which the antigen bound followed by elution with 0.2 M Glycine-HCl (pH 2.5) and concentration using an Amicon ultra-4 spin filter (Merck Millipore, Billerica, MA). Antibodies against phospho-p38 (#9211, 1:1,000), phospho-JNK1/2 (#9251, 1:1,000), and JNK1/2 (#9252, 1:1,000) were purchased from Cell Signaling Technology (Danvers, MA). IκBα (#sc-371, 1:1,000), p38 (#sc-535, 1:1,000), IRF3 (#sc-9082, 1:1,000) and HDAC1 (#sc-7872, 1:2,000) antibodies were obtained from Santa Cruz Biotechnology (Santa Cruz, CA). The anti-TUT4 (#18980-1-AP, 1:1,000) and GAPDH (#GTX627408, 1:5000) antibodies were obtained from Proteintech (Chicago, IL) and GeneTex (Irvine, CA), respectively. The IL-1β (#AF-401-NA, 1:2,000) antibodies and SP600125 (JNK inhibitor, #ab120065) were purchased from Abcam (Cambridge, UK) and R&D Systems (Minneapolis, MN). Antibodies against Regnase-1 (1:2,000) and β-actin (1:2,000) were kindly gifted from Drs. Shizuo Akira (Osaka University, Japan) and Sheng-Chun Lee (National Taiwan University, Taiwan), respectively. The anti-α-tubulin (#T9026, 1:5,000) and Flag (#F3165, 1:10,000) antibodies, anti-Flag® M2 Affinity Gel (#A2220), Lipopolysaccharide (#L6529, LPS, *E. coli* 055: B5), actinomycin D (#A9451), MG132 (#C2211), BMS-345541 (IKK inhibitor, #B9935), and DMSO (#D2650) were purchased from Sigma-Aldrich (St. Louis, MO). Macrophage colony-stimulating factor (M-CSF, #315-02) was from Peprotech (Rocky Hill, NJ). Puromycin (#P-600) and G-418 (#G-418) were purchased from Gold Biotechnology (St. Louis, MO). Poly(I:C) (#27-4732-01) and protein A-conjugated Sepharose (#17-0963-03) were obtained from GE Healthcare (Little Chalfont, UK). SB202190 (p38 inhibitor, #559388) and PD98059 (ENK inhibitor, #513001) were purchased from CalBiochem. R848 (#ALX-420-038-M025) was obtained from Enzo Life Science. CpG-1826 (#tlrl-1826) was obtained from Invivogen. SYBR Green II (#S7564) was obtained from Invitrogen (Carlsbad, CA). The secondary antibodies, Peroxidase-conjugated AffiniPure Goat Anti-Rabbit IgG (H + L, Cat.

111-035-003) and Peroxidase-conjugated AffiniPure Goat Anti-Mouse IgG (H + L, Cat. 115-035-003), were purchased from Jackson Immunoresearch (West Grove, PA).

**Mice**. *Tut7*$^{-/-}$ mice were generated by Clustered regularly interspaced short palindromic repeats (CRISPR)/Cas9-mediated genomic editing system in the Transgenic Mouse Models Core Facility of the National Core Facility Program for Biotechnology. *Tut7*$^{+/-}$ mice were generated by targeting two sgRNAs to the first and 26th introns of the *Tut7* gene on C57BL/6 genetic background. All animals were bred in the specific-pathogen-free animal facility. Mouse experiments were carried out in accordance with animal welfare guidelines and were approved by the Institutional Animal Care and Use Committee (IACUC) of the College of Medicine, National Taiwan University (approval no. 20150507).

**Cell cultures and preparation of bone marrow-derived macrophages**. RAW 264.7 macrophages were cultured in RPMI 1640 (Gibco, Pittsburgh, PA) supplemented with 10% (vol/vol) heat-inactivated fetal bovine serum (FBS) and 100 U/ml penicillin/streptomycin. HEK293T and HEK 293 cells were cultured in DMEM containing 10% FBS and 100 U/ml penicillin/streptomycin. All cells were maintained at 37 °C in a humidified 5% $CO_2$ atmosphere. BMDMs were prepared as previous description[5]. Briefly, femurs and tibia bones were collected from 6 to 8-week-old mice and bone marrow was flushed out with DMEM medium using a 25-gauge syringe. The bone marrow progenitor cells were harvested, and differentiated in high-glucose DMEM medium containing 20% L929 cell-conditioned medium for 7 days. Adherent BMDMs were collected and cultured in DMEM containing 10 ng/ml M-CSF overnight for further experiments.

**RNA extraction and Real-time quantitative RT-PCR**. Total cellular RNA was isolated using NucleoZol (MACHEREY-NAGEL, Düren, Germany). One μg of total RNA was used to synthesize cDNA with the RevertAid H Minus First-Strand cDNA Synthesis Kit (Thermo Scientific, Rockford, IL) following the manufacturer's instructions. The amount of cDNA was determined by Real-time quantitative PCR (RT-qPCR) using Maxima® SYBR Green/Fluorescein qPCR Master Mix (#4367659, Thermo Scientific, Rockford, IL) according to the manufacturer's instructions. All RT-qPCR values of interesting genes were normalized to cyclophilin A transcript as an internal control. All data were presented as fold-change relative to the unstimulated sample. The primer sequences are listed in Supplementary Table 5.

**shRNA-based gene silencing and lentiviral infection**. Lentiviral short hairpin RNA (shRNA) constructs expressing shRNAs against mouse TUT7, TUT4, and Regnase-1 in pLKO-puro vector were obtained from the National RNAi Core Facility Platform at the Institute of Molecular Biology/Genomic Research Center, Academia Sinica, Taiwan. The target sequences are
5′-ATGACAGGTGCTGCCGAATTT-3′ for *Tut7* shRNA-a,
5′-CTGAGTTCTTCTACGAATTTA-3′ for *Tut7* shRNA-b,
5′-AGCGAGGCCACACGATATTA-3′ for *Zc3h12a* shRNA-a,
5′-TATGGAATCAAGTGCCGATTT-3′ for *Zc3h12a* shRNA-b,
5′-GACAAACCGATTTCGAGAAAT-3′ for *Tut4* shRNA-a and
5′-ACACGTTTAGATAGCTTATTT-3′ for *Tut4* shRNA-b,
HEK293T cells were co-transfected with pLKO-shRNA, pMD.G and pCMVR8.91 plasmids using Turbofect (Fermentas, Schwerte, Germany) according to the manufacturer's recommendations. Cultural medium containing the lentivirus was harvested 48 and 72 h after transfection. RAW 264.7 macrophages were infected overnight with lentiviruses in the presence of 8 μg/ml polybrene (Sigma-Aldrich, St. Louis, MO) to promote lentiviruses attachment to cells followed by culturing in fresh medium for another 24 h. The infected cells were selected in 5 μg/ml puromycin-containing medium until the uninfected cells completely eliminated. The stable colonies were pooled for further experiments. *Tut7*$^{-/-}$ bone marrow progenitors were infected with lentiviruses expressing *Zc3h12a* shRNA on days 1 and 2 of differentiation into macrophages as mentioned previously[52]. After 4 days, cell medium was changed to fresh L-929-conditioned medium and the cells were differentiated for 3 more days.

**mRNA stability assay**. RAW 264.7 macrophages were challenged with 100 ng/ml LPS for 4 h, and then incubated with actinomycin D (5 μg/ml) for the indicated times. The cells were collected and total cellular RNAs were prepared for mRNA quantification by RT-qPCR as described above.

**Immunoblotting and immunoprecipitation**. Cells were collected and lysed in cold All Purpose Buffer (APB, 0.05 M Tris, pH 7.5, 0.25 M NaCl, 3 mM EDTA, 3 mM EGTA, 1% Triton X-100, 0.5% NP-40, 1% glycerol, 20 mM NaF, 40 mM β-glycerophosphate, 2 mM DTT, 1 mM phenylmethylsulfonyl fluoride (PMSF), 2 mM p-nitrophenyl phosphate (p-NPP), 1 mM sodium orthovanadate, and protease inhibitor cocktail including 2 μg/ml aprotinin, 1 μg/ml benzamidine hydrochloride hydrate, 1 μg/ml pepstatin A, and 2 μg/ml leupeptins) followed by incubation on ice for 15 min. The cell lysates were harvested after centrifugation of cell mixtures, and protein concentrations were determined by the Bradford assay (Bio-Rad, Hercules, CA) according to the manufacturer's instructions. Cellular extracts in

400 µl APB buffer were incubated with the appropriate antibody and Protein A-conjugated agarose beads overnight at 4 °C. The immunocomplexes were pelleted by centrifugation, washed three times with lysis buffer, and resuspended in SDS-polyacrylamide gel electrophoresis (SDS-PAGE) sample-loading buffer (50 mM Tris-HCl pH 6.8, 10% glycerol, 2% SDS, 1.25% β-mercaptoethanol, and 0.1% bromophenol blue). Protein samples were separated by SDS-PAGE, and transferred to the PVDF membranes. The PVDF membranes were incubated with the indicated primary antibody overnight at 4 °C, followed by incubation of HRP-conjugated secondary antibody for 1 h at 25 °C. The immunoreactive signals were detected using Western Lighting® Plus-ECL (PerkinElmer, Waltham, MA) based on the manufacturer's instructions.

**Dual-luciferase reporter and enzyme-linked immunosorbent assays**. Cells were co-transfected with the indicated firefly luciferase reporter plasmid, pRL-TK-renilla luciferase plasmid, and TUT7 or empty control plasmid, and cultured for 48 h. Cells were harvested and lysed, and firefly and renilla luciferase activities were determined using the Dual-Luciferase reporter assay system (Promega, Madison, WI) according to the manufacturer's instructions. The levels of cytokines in cultural supernatants were measured by DuoSet ELISA systems (R&D Systems, Minneapolis, MN) following the manufacturer's instructions.

**In vitro transcription and In vitro uridylation**. In vitro transcription was carried out by T7 RNA polymerase (Promega, Madison, WI) according to the manufacturer's instructions. Briefly, 5 µg of linearized DNA template was incubated with 40 units T7 RNA polymerase in 100 µl reaction mixtures containing 40 mM Tris, pH 7.9, 6 mM $MgCl_2$, 2 mM spermidine, 10 mM NaCl, 10 mM DTT, 2.5 mM NTP mix, 100 units RNase inhibitor, for 2 h at 37 °C. RNA transcripts were then purified by NucleoZol, precipitated with 100% isopropanol (Sigma-Aldrich, St. Louis, MO), and resolved in 10 µl RNase-free water. For in vitro uridylation assay, 1 µg in vitro synthesized RNAs were incubated with wild-type or enzyme-dead TUT7 immunoprecipitated from HEK293T cells transfected with TUT7 expression plasmids and α-$^{32}$p-UTP for 1 h at 37 °C. RNA samples were separated on a 1% formaldehyde-agarose gel electrophoresis and visualized by SYBR Green II staining and radiography.

**RNA-immunoprecipitation**. Cells were harvested and lysed in APB lysis buffer containing protease inhibitors and RNase inhibitor. 5% of cell lysates were kept as the inputs while remaining lysates were incubated with the appropriate antibody and Protein A-conjugated agarose beads for 4 h at 4 °C. The beads were pelleted and washed three times with APB containing RNase inhibitor. 5% of the beads were subjected to SDS-PAGE to validate the efficiency of the immunoprecipitation. The remaining beads and the inputs were resuspended in 200 µl NucleoZol for RNA extraction followed by RT-PCR and RT-qPCR as mentioned above. The primer sequences, Zc3h12a_3′-UTR Forward 5′-ATCACAGATAGCGGTCCCCA-3′, Zc3h12a_3′-UTR Reverse 5′-GGCAATAGCTTTTTTTTTCTTTTAA-3′, Il6 Forward 5′-ACAAGAAAGACAAAGCCAGAGTC-3′, and Il6 Reverse 5′-ATTG GAAATTGGGGTAGGAAG-3′, were used in RT-PCR and RT-qPCR.

**RNA sequencing**. Total RNAs harvested from BMDMs were purified by Trizol and subjected to next-generation sequencing and data analysis (Changgong Biotechnology, Taipei, Taiwan) following the manufacturer's instructions. Briefly, the total RNAs were isolated by poly-dT magnetic beads, fragmented, reverse transcribed using random primers, and synthesized the second strand using linkers containing barcoded to generate the cDNA libraries. The cDNA libraries were amplified by PCR for the indicated cycles and the specific size products were selected and purified before sequencing on an Illumina HiSeq PE150 for 75 nts of each read. Total of 4.45–11.8 million reads (Reads Per Kilo bases per Million reads, RPKMs) were obtained from each sample. RPKMs were compared to mouse RefSeq-RNA mm10 and quantitated using FANSe3 (Fast and Accurate mapping algorithm for Next-generation Sequencing, the 3rd generation[53,54]). The differential genes expression between control and LPS stimulation of each genotype (wild-type and $Tut7^{-/-}$) was quantified using edgeR[55] and FANSe3. To identify the differential genes expression in response to LPS, fold-change (>1.2X) and t tests (p value < 0.05) of $Tut7^{-/-}$ BMDMs at 4 h after LPS treatment were compared to that of $Tut7^{+/+}$ BMDMs. All detailed analysis was consulted by Next-generation Sequencing Analysis Cloud System (Chi-Cloud). The differentially expressed innate immune-related genes are listed in Supplementary Table 2 (downregulated genes) and Supplementary Table 3 (upregulated genes). The NCBI GEO accession number for this experiment in present paper is GSE136161.

**TAIL-Seq of specific genes**. Total RNAs collected from BMDMs were purified by NucleoZol and subjected to 3′-linker ligation with adenylated linker-1, rRNA depletion, and cDNA library construction. Briefly, linker-1 (CTGTAGGCACCA TCAAT) was firstly adenylated with Mth RNA ligase (#M2611A, New England BioLabs, Ipswich, MA), and adenylated linker-1 was ligated to the extracted RNAs. The 3′-linker-ligated RNAs were further depleted of rRNA by NEBNext® rRNA Depletion Kit (E6310S, New England BioLabs, Ipswich, MA) and fragmented by

RNase T1 (EN0541, Thermo Scientific, Rockford, IL). Ribosomal RNA-free transcripts were reversely transcribed using SuperScript™ III First-Strand Synthesis System (Thermo Scientific, Rockford, IL) with primer complement to linker-1 with partial illumina sequence (5′-ACACTCTTTCCCTACACGACGCTCTTCCGATC TATTGATGGTGCCTACAG-3′). The 3′ end of individual transcript was amplified with gene specific primer and primer complement to linker-1 (TTTAAATGAAA AAGGTTGACAAAATAAA for Zc3h12a-forward primer and TGTTTTAGACTGT CTTCAAACAAATAAA for Il6-forward primer, and RT linker-1 for reverse primer). Then the second-run PCR was used to add all the necessary sequences that were required for illumine sequencing using Q5® High-Fidelity DNA Polymerase (M0491S, New England BioLabs, Ipswich, MA). The amplified libraries were purified using 1X Sera-Mag Select magnetic beads (29343045, Cytiva/GE Healthcare, Little Chalfont, UK). The specific size (200–400 bps) products were selected, purified, and mixed with 20% (of total libraries) PhiX control library before sequencing on an Illumina HiSeq 2500 Rapid v2, Paired End 2*150 for 150 nts of each read. Total of 0.3–1.4 million reads were obtained from each sample. The reads were mapped to mouse RefSeq-RNA mm10 using HISAT2, and further analysis of Zc3h12a and Il6 was trimmed by 5′ and 3′ Illumina adapter sequences and filtered using Filter FASTA program to filter out Zc3h12a and Il6 by linker-1 sequence and the specific criteria (TTTAAATGAAAAAGGTTGACAAAATAAA for Zc3h12a and TGTTTAGACTGTCTTCAAACAAATAAA for Il6) on Galaxy bioinformatics online tool (https://usegalaxy.org/). The NCBI GEO accession number for this experiment in present paper is GSE164259. The oligonucleotides used in TAIL-Seq were listed in Supplementary Table 6.

**RNA electrophoretic mobility shift assay and competition assay**. Cy3-labeled and non-labeled RNAs listed in Supplementary Table 7 were purchased from Genomics (New Taipei City, Taiwan). Seventy five fmole synthesized RNAs were incubated with immunopurified TUT7 from HEK293T cells at 37 °C for 30 min, and RNA samples were then mixed with RNA loading dye (1× TBE, 20% glycerol, 0.05% bromophenol blue) and separated on a 6% native PAGE, followed by detection of fluorescence signals from Cy3-fluorophore or SYBR Green II staining using the iBright FL1000 Imaging System (Invitrogen, Carlsbad, CA).

**Sequence alignment of TUT7 among different species**. The human TUT7 orthologous proteins in 9 different species were aligned using the Clustal Omega program (Version 1.2.1) provided by EMBL-EBI (http://www.ebi.ac.uk/). The accession numbers for TUT7 proteins of different species are as follows: Homo sapiens (H. sapiens, NP_078893), Pan troglodytes (P. troglodytes, XP_001138296), Bos Taurus (B. Taurus, NP_001192681), Mus musculus (M. musculus, NP_705766), Rattus norvegicus (R. norvegicus, XP_006253595), Gallus gallus (G. gallus, NP_001258876), Xenopus tropicalis (X. tropicalis, XP_002935025), Danio rerio (D. rerio, XP_009300328), Caenorhabditis elegans (C. elegans, NP_498099), and Schizosaccharomyces pombe (S. pombe, NP_594901), all of which were adopted from the NCBI database (http://www.ncbi.nlm.nih.gov/).

**Cytosolic and nuclear fractionation**. RAW 264.7 cells stimulated with or without 100 ng/ml LPS were washed once with cold PBS and lysed in 100-200 µl nuclear fractionation buffer A (10 mM HEPES, 1.5 mM $MgCl_2$, 10 mM KCl, 0.5 mM DTT, 0.05% NP-40, pH 7.9, 1 mM PMSF, 2 mM p-NPP, 1 mM $Na_3VO_4$, and 1× protease inhibitor cocktail). The cells were left on ice for 15 min, and then pelleted at 3,500 xg for 10 min at 4 °C. The supernatants were recovered as the cytosolic fraction. The pellets were then washed twice with 200 µl Nuclear Fractionation Buffer A and resuspended in 50-80 µl nuclear fractionation buffer B (5 mM HEPES, 1.5 mM $MgCl_2$, 300 mM NaCl, 0.2 mM EDTA, 0.5 mM DTT, 26% glycerol, pH 7.9, 1 mM PMSF, 2 mM p-NPP, 1 mM $Na_3VO_4$, 1X protease inhibitor cocktail) followed by sonication (Branson sonifer 205) using three 30 s bursts separated by 1 min intervals and incubation on ice for 30 min. The lysates were pelleted at 21,000 xg for 20 min at 4 °C and the supernatants were collected as the nuclear fraction.

**The endotoxemia mouse model**. 8-week-old male $Tut7^{+/+}$ and $Tut7^{-/-}$ mice were injected intraperitoneally with 50 µg/kg LPS. Whole blood samples were collected at 2 h after LPS challenge, and cytokines in sera were measured using ELISA.

**Statistical analysis**. The results are presented as mean ± SD. The differences between two groups were determined by two-tailed Student's t test. The exact p values are shown in the figures if p is <0.05, which is considered statistically significant.

## Data availability

RNA-Seq and TAIL-Seq data have been deposited into NCBI GEO under the accession number GSE136161 and GSE164259, respectively. The datasets of LPS-induced BMDMs and RAW 264.7 cells were obtained from GEO with the accession number of GDS5623. The data that support the findings of this study are available from the corresponding author upon reasonable request. Source data are provided with this paper.

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

## Acknowledgements

We thank Drs. Shizuo Akira, Keith L. Kirkwood, Joseph P. Mizgerd, Sheng-Chun Lee, and Narry V. Kim for gifts of plasmids and antibodies; the Transgenic Mouse Model Core Facility of the National Core Facility Program for Biotechnology; the Ministry of Science and Technology, Taiwan; and the Gene Knockout Mouse Core Laboratory of National Taiwan University Center of Genomic Medicine for providing technical services. We are grateful to Dr. Betty An-Ye Wu-Hsieh for invaluable discussions and critical reading of the paper. We thank the staff of the Second Core Lab, Department of Medical Research, National Taiwan University Hospital for technical support during the study. We are

grateful for the technical support provided by the 3rd Core Facility and the Microscopy Core Facility of Department of Medical Research at National Taiwan University Hospital. We are grateful to the High Throughput Genomics Core Facility, Biodiversity Research Center, Academia Sinica for the technical support of TAIL-Seq. This work was supported by the Ministry of Education in Taiwan, National Taiwan University (110L901402B to L.-C.H.), the Ministry of Science and Technology (MOST) of Taiwan (105-2320-B-002-060-MY3, 108-2320-B-002-020-MY3 and 110-2634-F-002-044 to L.-C.H, and 109-2314-B-002-080 to C.-Y.L.), National Health Research Institutes, Taiwan (NHRI-EX106-10630SI and NHRI-EX110-11031SI to L.-C.H.), and Excellent Translational Medicine Research Projects of National Taiwan University College of Medicine and National Taiwan University Hospital (NSC-145-62 and 110C101-061). RNAi reagents were obtained from the National RNAi Core Facility supported by the National Core Facility Program for Biotechnology Grants of MOST (100-2319-B-001-002).

## Author contributions

C.-C.L., Y.-R.S., and L.-C.H., designed the research. C.-C.L. and Y.-R.S. performed the experiments. I.-S.Y. generated the $Tut7^{-/-}$ mice. C.-C.L., Y.-R.S., L.-C.H., and H.-Y.T. analyzed the data. S.-Y.G., C.-C.C., Y.-Y.Y., T.-Y.L., and C.-Y.L. assisted with some experiments. C.-H.C. generated TUT7 antibody. C.-C.L. and L.-C.H. wrote the paper. All authors discussed the results and approved the paper.

## Competing interests

The authors declare no competing interests.
