## [Peer Review File · Nature Communications]

Reviewers' comments:

Reviewer #1 (Remarks to the Author):

In their manuscript NCOMMS-19-26858 entitled "Terminal uridylyltransferase 7 regulates TLR4-triggered inflammation by controlling Regnase-1 mRNA uridylation and degradation" Lin and colleagues show that Tut7 is induced by LPS stimulation. Upon silencing or knockout of Tut7, they find decreased LPS-dependent induction of IL-6 encoding mRNAs, as well as other mRNAs, which are known targets of Regnase-1. Mechanistically, the downregulation of IL6 was sensitive to the catalytic activity of Tut7, occurred post-transcriptionally and required cis-elements in the IL6 3'UTR that are targeted by Regnase-1. Tut7 did not bind to the IL6 mRNA but to the Zc3h12 mRNA and required Regnase-1 expression to exert its function. In fact, Regnase-1 appeared to be upregulated in the absence of Tut7, and Tut7 was able to uridylate the Zc3h12 mRNA in vitro and potentially also in cells. The authors involve Dis3L2 as additional downstream effector of Zc3h12a mRNA decay, however, the functional evidence is rather limited. Altogether, the paper promotes a very intriguing concept with biological relevance, but appears rather preliminary in its mechanistic analyses.

Major points

1. The authors should use the HUGO nomenclature for naming RNAs and genes as well as knockouts (i.e. Zc3h12a and Zcchc11)
2. How can TLR4 but not the other TLRs (Fig. S1a) induce Tut7 through the induction of NF-kappaB (as shown in Fig. 8), if NFkappaB is also induced by the other TLRs and many other stimuli?
3. The authors investigate at great length how Tut7 regulates IL-6 through its 3'-UTR, although this just recapitulates what has been well established for Regnase-1 before (Fig.2-3). In Fig. 4 they then recapitulate how the 3' UTR of Zc3h12a responds to Tut7 i.e. in a similar way as it responds to Regnase-1. Therefore, most of the data in Fig. 2-4 are not that informative and in many cases only involve overexpression experiments that then result in rather small effects (typically less or much less than two-fold). Figs. 2-4 could be condensed into one Figure, which shows just the physiologic relevant experiments. Also, the original literature (3'UTR mutagenesis of Il6 and Zc3h12a mRNAs) should be cited appropriately.
4. Why is Zc3h12a mRNA not upregulated in Fig.1g? This is contradictory to Fig. 5b!
5. Figure 5d provides nice proof, that Tut7 can uridylate the Zc3h12a mRNA, dependent on its 3'UTR. However, it is not clear how it recognizes the 3'UTR and even the specific stem-loop in it, (i.e. the question is whether Tut7 itself recognizes the stem-loop structure or associates with Regnase-1 to bind it). The authors should also repeat this experiment with Regnase-1 Ko cells to address this point. Fig. 5d should also test the Il6 3' UTR.
6. The binding preference of Tut7 has not been analyzed at all. RNA-EMSA would be very important to understand the preference of certain but not other stem-loop structures.
7. The cloning and sequencing of Zc3h12a 3'-UTR sequences (Fig. 6c) has few sequences and few uridylation events. Unfortunately, blasting the first sequence shown in Fig.6c does not yield in Zc3h12a sequences but in a synthetic construct (cloning vector pJPVCS) suggesting some contamination in this experiment. The authors should replace these data with a real TAIL-Seq experiment and perform sophisticated bioinformatic analyses.
8. The colocalization and functional data in Figure 7 are not trustworthy. Fig. 7a just shows diffuse localization of Tut7 and Dis3L2 in the cytoplasm, Fig. 7b only proves that Tut7 is not localized to P bodies, and the knockdown approaches give marginal effects on IL6 and on Zc3h12a mRNA levels, which are not even negatively correlated in the both shRNAs (i.e. sh-a has a smaller effect on Il6 and bigger effects on Zc3h12a, while sh-b downregulates IL6 better, but does not regulate Zc3h12a more effectively). Western blots would be required to demonstrate upregulation of Regnase-1 in sh-Dis3L2 knockdown cells.
9. The manuscript would also require some more language editing, for example: Among the 352 genes, 111 of them involving in innate immune response were...

Reviewer #2 (Remarks to the Author):

Lin and colleagues report in this manuscript that, during TLR4-triggered inflammation, terminal nucleotidyltransferase 7 (TUT7) is induced and required for the production of some key cytokines including IL-6. Mechanism-wise, they found that TUT7 binds to a stem-loop in the Regnase 1 mRNA to uridylate and destabilize the Regnase 1 mRNA. The RNase Regnase-1 in turn recognizes a stem-loop in the IL-6 mRNA 3' UTR to destabilize the mRNA.

Overall this is an elegantly performed study with solid evidence and interesting new mechanism for gene regulation in inflammation. Prior to publication, the following issues should be addressed though:

1. Figure 6 and Supplementary Figure 8 present the 3' RACE data for regnase-1 and il-6 mRNA, respectively. Why do the 3' UTR sequences vary between the clones and why poly(A) tail is absent in most clones? Please provide an explanation about the results.
2. Please also specify the primer used for cDNA synthesis for RT-PCR experiments, oligo dT or random primer? If the mRNAs indeed lack poly(A) tails prior to LPS treatment, the RT-PCR experiments should be done with random primers.

Reviewer #3 (Remarks to the Author):

A growing body of evidence implicates post-transcriptional processes in the induction of cytokine synthesis in inflammation. In particular, previous studies have established that the ribonuclease Regnase-1 regulates IL6 mRNA stability. The authors here present data that collectively give rise to a model in which Regnase-1 is in turn regulated by the terminal uridyl transferase TUT7, which according to their model negatively regulates Regnase-1 mRNA by binding to a stem-loop in its 3' UTR, uridylating the mRNA 3' end and hence targeting it for destruction by the Dis3L2 exoribonuclease. As the authors show TUT7 is induced by TLR4 engagement, the proposed pathway could help to explain how pro-inflammatory signals cause increased synthesis of cytokines such as IL6. Other studies have focused on the capacity of TUT7 and, more particularly, the related uridyl transferase TUT4 (ZCCHC11) to target IL6-regulatory micro-RNAs; this study is the first, to my knowledge, to implicate TUT7 in the control of the mRNA level of an IL6 regulator.

Some aspects of the study provide evidence to support the authors conclusions, though the effect sizes, for example in terms of the impact of experimental modulation of TUT7 on cytokine production, are generally quite modest. In particular, the induction of TUT7 expression by LPS treatment of a macrophage cell line is well documented, as are the structure-function experiments indicating that TUT7 influences IL6 mRNA through the latter's conserved stem-loop structure. Given the known involvement of Regnase-1 in modulating IL6 mRNA through this stem-loop, the follow-up experiments on the Regnase-1 mRNA 3'UTR represent a logical progression and support the idea that TUT7 level is somehow influencing Regnase-1, directly or indirectly, through its 3'UTR and potentially through a stem-loop structure. The RNA-IP data are not sufficient in themselves to substantiate a direct interaction between TUT7 and the Regnase-1 3'UTR, though they could be consistent with such an interaction. The in vitro uridylation assay using synthetic mRNA substrates is intriguing and the resulting data (Fig. 5d) could suggest selective uridylation of the Regnase-1 mRNA 3' end by TUT7, though numerous details and control data are missing. Do the synthetic RNA substrates carry a poly(A) tail, as they would in vivo, and is it clear that each of the RNA substrates used is capable of 3' end modification by a control nucleotidyl transferase such as poly(A) polymerase? Are accessory factors required for the apparent interaction between TUT7 and the Regnase-1 3'UTR, by analogy with

TUT4 binding to pre-miRNAs?

In summary, I find this an intriguing and potentially valuable study, though the key question of how TLR4 engagement leads to TUT7 induction is not addressed. In addition, the following points in my view render the manuscript unsuitable for publication in its current form.

1. The 3' RACE data in Figure 6c are deeply puzzling. The cartoon at the top of the figure part suggests that the sequences shown are downstream from the mRNA 3' end – in other words, the cleavage/polyadenylation site, so it would be expected that the sequences would be predominantly poly(A), with perhaps additional untemplated Ts at the 3' end. Instead, many of the sequences do not contain poly(A) tracts; I assume the intention is instead to show for each mRNA the hexanucleotide poly(A) signal, highlighted in yellow, and all sequences downstream from this, in other words the 3' end of the 3' UTR plus any non-templated nucleotides. But the sequence highlighted in yellow is in numerous instances AAATAA, which unlike the canonical AATAAA would not be expected to act as a functional poly(A) signal. Authentic mRNA sequences would be expected to have 15-30 nt of templated nucleotides downstream from this signal before the cleavage site and poly(A) tail, but this is seen only in a minority of the sequences shown. More worryingly still, the non-poly(A) sequences are mostly different from each other, contain unexplained gaps and do not obviously correspond to the 3' UTR of Regnase-1. For example, BLAST indicates that the first sequence has no significant match to the murine or human genome, but a highly significant 44/44 match to a number of bacterial cloning vectors; similarly, the single transcript with an untemplated U in the control TUT7+/+ sample shows no significant match to the murine or human genome, but is highly similar to a bacterial 16S rRNA sequence. Given this, it is by no means clear how the final T in this sequence has been designated as 'untemplated'. In my view no conclusions may be drawn from these data. As a consequence, the authors' conclusions about the role of TUT7 in the uridylation of Regnase-1 mRNA, and the relationship between this modification and poly(A) tail length are wholly unsound. Similarly, the first sequence shown in Supplemental Figure 8B seems to be derived not from the IL6 gene, as claimed, but from a bacterial cloning vector. I have not taken the time to identify the remaining sequences, but this is a fairly basic part of the bioinformatic analysis that should have been carried out by the authors. Unfortunately, none of the 3'RACE data presented would appear to be informative.

2. The immunofluorescence co-localisation data presented in Figure 7 are of marginal value; as both TUT7 and DIS3L2 are both known to be cytoplasmic proteins, their apparent co-localisation adds little additional information.

Point-by-point response to reviewer's comments:

Editor:

We consider it particularly important that the conclusions be supported by additional experimentations that should include: TAIL-Seq experiment to show tail sequence change of Regnase-1 and IL6 3'-UTR, and test of Regnase-1 Ko cell and IL6 3' UTR in fig 5d. We feel that Dis3l2 part (Fig 7) could be removed if not supported.

Our response:

We appreciate the opportunity to revise our manuscript. All of the reviewers' comments are addressed in this revision. In response to the comments and suggestions by the editor and reviewers #1 and #3, we performed TAIL-Seq to assess the impact of TUT7 on uridylation of *Zc3h12a* and *Il6* 3' ends after LPS challenge. The results demonstrate that TUT7 mediates *Zc3h12a*, but not *Il6*, uridylation in response to LPS. The data are shown in Fig. 6 and Supplementary Fig. 9. We apologize for the confusion generated from the original Fig. 5d. The TUT7 protein used in *in vitro* uridylation was immunopurified from HEK293T cells transfected with Flag-TUT7 plasmid. Regnase-1 is an inducible molecule and its protein level is very low in HEK293T cells as shown in Supplementary Fig. 8a. We did not detect Regnase-1 in the immunocomplex of TUT7 by immunoblotting (Supplementary Fig. 8b). Following the suggestions of the editor and reviewers #1 and #3, we removed the results regarding Dis3l2 in the revised manuscript.

Reviewer #1:

Altogether, the paper promotes a very intriguing concept with biological relevance, but appears rather preliminary in its mechanistic analyses.

Our response:

We thank the reviewer for his/her helpful comments and suggestions. The reviewer's comments are addressed as follows:

1. The authors should use the HUGO nomenclature for naming RNAs and genes as well as knockouts (i.e. *Zc3h12a* and *Zcchc11*)

Our response:

All RNAs and genes are named according to the HUGO nomenclature in the revised manuscript. Of note, the HUGO nomenclature for *Zcchc11* is *Tut4*, so we used *Tut4* instead of *Zcchc11* in the revised manuscript.

2. How can TLR4 but not the other TLRs (Fig. S1a) induce *Tut7* through the

induction of NF-kappaB (as shown in Fig. 8), if NFkappaB is also induced by the other TLRs and many other stimuli?

Our response:

We apologize for the confusion generated from the original Fig. 8. We did investigate the underlying mechanism by which TLR4 induces TUT7, but only summarized the findings without presenting the original results in the original manuscript. We now include these results in the revised manuscript (Supplementary Fig. 1a). Our results clearly reveal that IKK, but not MAPK JNK nor ERK, is required for TLR4-stimulated TUT7 expression. Extending this investigation to include other TLR ligands, we found that IKK, but not JNK, activation is also needed for TUT7 induction by TLR1/2 and TLR7/8 ligands (Supplementary Fig. 1c). Thus, it is possible that induction of TUT7 by different TLR ligands is through the IKK/NF- κ B pathway. However, since we only focus how TUT7 regulates TLR4-driven IL-6 expression in this study, we present data of TLR4, but not other TLRs, inducing TUT7 through the activation of IKK-NF- κ B in Fig. 7 (the original Fig. 8).

3. The authors investigate at great length how Tut7 regulates IL-6 through its 3'-UTR, although this just recapitulates what has been well established for Regnase-1 before (Fig.2-3). In Fig. 4 they then recapitulate how the 3' UTR of Zc3h12a responds to Tut7 i.e. in a similar way as it responds to Regnase-1. Therefore, most of the data in Fig. 2-4 are not that informative and in many cases only involve overexpression experiments that then result in rather small effects (typically less or much less than two-fold). Figs. 2-4 could be condensed into one Figure, which shows just the physiologic relevant experiments. Also, the original literature (3'UTR mutagenesis of Il6 and Zc3h12a mRNAs) should be cited appropriately.

Our response:

We thank the reviewer's suggestion. After thorough consideration and discussion, we prefer to maintain the current format for the reasons listed below:

- (1) Fig. 2 is to determine how and through which cis-element TUT7 modulates IL-6 mRNA induced by LPS. Though the data are lengthy, they clearly ruled out the involvement of ARE-binding proteins and miRNAs in TUT7-mediated IL-6 expression induced by LPS, and eventually helped us to narrow down our target to Regnase-1 in the Fig. 3.
- (2) Fig. 3 is to demonstrate TUT7 regulates LPS-induced IL-6 expression through Regnase-1, and this effect is at work in both RAW264.7 cell line

and primary bone marrow macrophages.

- (3) To logically present data to show regulation through TUT7-Reganase-1-IL-6 axis during TLR4 activation without confusing the readers, it is necessary to separate Fig. 2 and Fig. 3.
- (4) Fig. 4 is to characterize how TUT7 modulates *Zc3h12a* expression, and to our surprise, it is through recognizing the stem-loop secondary structure on *Zc3h12a* 3'-UTR. It remains to be determined how TUT7 recognizes its substrates even though its various physiological functions have been studied extensively. Therefore, we believe that the information generated from Fig. 4 should be helpful for the determination of the recognition motif for TUT7 in the future.

We apologize for the inappropriate citation for 3'UTR mutagenesis of *Il6* and *Zc3h12a* mRNAs. We have made corrections and correctly cited the original publications in the revised manuscript.

4. Why is *Zc3h12a* mRNA not upregulated in Fig. 1g? This is contradictory to Fig. 5b!

Our response:

We apologize for the confusion generated from Fig. 1g. Fig. 1g shows only the regulated genes involved in innate immune response in the Innate DB database we used for RNA-seq analysis, and *Zc3h12a* is not included. In addition, data in Fig. 1g were generated from BMDMs treated with LPS for 4 hr. *Zc3h12a* mRNA expression peaked at 1-2 hr post-LPS treatment (Fig. 3e) and gradually declined thereafter. To assess *Zc3h12a* induction by LPS, we harvested cells at time point 2 hr post-LPS treatment. Data presented in Fig. 5b were from 2 hr time point. However, we did go back to examine our raw data for Fig. 1g. *Zc3h12a* was indeed slightly increased in *Tut7*^{-/-} macrophages at 4 hr after LPS stimulation compared to wild type cells [duplicate samples, average 66 (wild type) vs. 71 (*Tut7*^{-/-}) rpk], whereas no difference in *Zc3h12a* expression was observed in non-treated control and *Tut7*^{-/-} cells (13.7 vs 13.8 rpk). As a common practice, results obtained from RNA-Seq and TAIL-Seq are needed to be confirmed by a different approach, such as RT-qPCR. We show the RT-qPCR results below for the same RNA sample used for TAIL-Seq in Fig. 6. The data again confirm that *Zc3h12a* is upregulated in LPS-treated *Tut7*^{-/-} macrophages.

5. Figure 5d provides nice proof, that Tut7 can uridylylate the Zc3h12a mRNA, dependent on its 3'UTR. However, it is not clear how it recognizes the 3'UTR and even the specific stem-loop in it, (i.e. the question is whether Tut7 itself recognizes the stem-loop structure or associates with Regnase-1 to bind it). The authors should also repeat this experiment with Regnase-1 Ko cells to address this point. Fig. 5d should also test the Il6 3' UTR.

Our response:

We appreciate the reviewer's comment. Following a previous study from Dr. Narry Kim's lab (Lim J et al., Cell, 2014; PMID: 25480299), TUT7 protein we used in *in vitro* uridylation assay was immunopurified from HEK293T cells transfected with Flag-TUT7 plasmid. Regnase-1 is an inducible protein, and its protein level is low in resting cells (Matsushita K et al., Nature, 2009; PMID: 19322177). We confirmed that Regnase-1 protein in HEK293T cells is lower than resting RAW264.7 cells (Supplementary Fig. 8a). In addition, we did not detect Regnase-1 in the TUT7 immunocomplex by immunoblotting analysis (Supplementary Fig. 8b). Regnase-1 is known to bind to the stem-loop structure of both *Zc3h12a* and *Il6*. Our RNA EMSA analysis show that immunopurified TUT7 only bound to RNA probe containing *Zc3h12a* stem-loop, but not that containing *Il6* stem-loop (Fig. 5c). *In vitro* uridylation experiments with constructs containing different combinations of CDSs, 3'-UTRs, and the stem-loop structures from the 3'-UTR of *Il6* and *Zc3h12a* reveal that TUT7 only uridylylates transcripts containing *Zc3h12a* stem-loop, but not *Il6* (Fig. 5f-g). These data together indicate that TUT7 directly recognizes *Zc3h12a* stem-loop without involving Regnase-1.

6. The binding preference of Tut7 has not been analyzed at all. RNA-EMSA would be very important to understand the preference of certain but not other stem-loop structures.

Our response:

Following the reviewer's suggestion, we performed RNA-EMSA using chemically synthesized RNA probes containing *Il6* or *Zc3h12a* stem-loop. Results in Fig. 5c show that TUT7 only recognizes *Zc3h12a* stem-loop, but not *Il6* stem-loop.

7. The cloning and sequencing of *Zc3h12a* 3'-UTR sequences (Fig. 6c) has few sequences and few uridylation events. Unfortunately, blasting the first sequence shown in Fig.6c does not yield in *Zc3h12a* sequences but in a synthetic construct (cloning vector pJPVCS) suggesting some contamination in this experiment. The authors should replace these data with a real TAIL-Seq experiment and perform sophisticated bioinformatic analyses.

Our response:

Indeed because we did not double check all the sequences presented in the original Fig. 6c and Supplementary Fig. 8, thus did not detect contamination. We also lacked experience in preparing 3'-ligation RACE library. Following the editor and Reviewers' suggestion, we performed specific gene TAIL-Seq to deep-sequence the 3' ends of *Zc3h12a* and *Il6*. Our results demonstrate that LPS stimulation increased *Zc3h12a* uridylation, especially the percentage of oligo-uridylation (≥ 2 U), in wild-type BMDMs (Fig. 6b). Deleting *Tut7* reduced *Zc3h12a* oligo-uridylation (Fig. 6b). On the contrary, TUT7 has marginal impact on *Il6* uridylation response to LPS (Supplementary Fig. 9). In addition, *Zc3h12a* with oligo-uridine (≥ 2 U) on its 3' end possesses shorter length of poly(A) in wild type cells after LPS challenge, whereas there is no such correlation in *Tut7*^{-/-} cells (Fig. 6c). Interestingly, we also found two other non-A terminal modifications, cytosine and quinone, on 3' end of *Zc3h12a*, but neither one is modulated by TUT7 (Fig. 6d). These results confirm that TUT7 mediates *Zc3h12a* uridylation and controls its degradation.

8. The colocalization and functional data in Figure 7 are not trustworthy. Fig. 7a just shows diffuse localization of *Tut7* and *Dis3l2* in the cytoplasm, Fig. 7b only proves that *Tut7* is not localized to P bodies, and the knockdown approaches give marginal effects on *Il6* and on *Zc3h12a* mRNA levels, which are not even negatively correlated in the both shRNAs (i.e. sh-a has a smaller effect on *Il6* and bigger effects on *Zc3h12a*, while sh-b downregulates *Il6* better, but does not regulate *Zc3h12a* more effectively). Western blots would be required to demonstrate upregulation of *Regnase-1* in sh-*Dis3l2* knockdown cells.

Our response:

We appreciate the reviewer's suggestion. We are not able to obtain a good

antibody against Dis3L2 to improve our immunofluorescence staining. We also used different shRNA against *Dis3l2* but could not achieve good knockdown efficiency. We took the reviewer's suggestion and removed the figures concerning Dis3L2 from the revised manuscript.

9. The manuscript would also require some more language editing, for example: Among the 352 genes, 111 of them involving in innate immune response were...

Our response:

We appreciate the reviewer's suggestion. The revised manuscript has been edited by an English-proficient scientist. We also rephrased the sentence raised by the reviewer to "Among the 352 genes, 111 genes involving in the innate immune response were downregulated in *Tut7*^{-/-} cells." in Page 8 of the revised manuscript.

Reviewer #2:

Lin and colleagues report in this manuscript that, during TLR4-triggered inflammation, terminal nucleotidyltransferase 7 (TUT7) is induced and required for the production of some key cytokines including IL-6. Mechanism-wise, they found that TUT7 binds to a stem-loop in the Regnase 1 mRNA to uridylate and destabilize the Regnase 1 mRNA. The RNase Regnase-1 in turn recognizes a stem-loop in the IL-6 mRNA 3' UTR to destabilize the mRNA. Overall this is an elegantly performed study with solid evidence and interesting new mechanism for gene regulation in inflammation. Prior to publication, the following issues should be addressed though:

Our response:

We thank the reviewer for the support of our work.

1. Figure 6 and Supplementary Figure 8 present the 3' RACE data for regnase-1 and il-6 mRNA, respectively. Why do the 3' UTR sequences vary between the clones and why poly(A) tail is absent in most clones? Please provide an explanation about the results.

Our response:

It is well accepted that transcripts generated by RNA polymerase II undergo cleavage 15-30 nt downstream of a polyadenylation signal (PAS; usually AAUAAA sequence) in its 3' UTR, followed by poly(A) tail synthesis at the point of cleavage. The mature mRNA transcripts containing poly(A) tails and 5'-Cap

For RT-PCR analysis in this study, we used oligo(dT) to synthesize cDNA to avoid non-specific binding or binding to degraded RNA fragments. To clarify the point raised by the reviewer, we also synthesized cDNAs using either oligo(dT) or random primer followed by Real-time PCR analysis. The results as shown below clearly demonstrated that regardless of what type of primers we used for cDNA synthesis, there was a similar induction pattern of *Il6*, *Zc3h12a* and *Tnf* after stimulation by LPS.

Reviewer #3:

As the authors show TUT7 is induced by TLR4 engagement, the proposed pathway could help to explain how pro-inflammatory signals cause increased synthesis of cytokines such as IL6. Other studies have focused on the capacity of TUT7 and, more particularly, the related uridyl transferase TUT4 (ZCCHC11) to target IL6-regulatory micro-RNAs; this study is the first, to my knowledge, to implicate TUT7 in the control of the mRNA level of an IL6 regulator.

Our response:

We appreciate the reviewer for the support of our work.

Some aspects of the study provide evidence to support the authors conclusions, though the effect sizes, for example in terms of the impact of experimental modulation of TUT7 on cytokine production, are generally quite modest. In particular, the induction of TUT7 expression by LPS treatment of a macrophage cell line is well documented, as are the structure-function experiments indicating that TUT7 influences IL6 mRNA through the latter's conserved stem-loop structure. Given the known involvement of Regnase-1 in modulating IL6 mRNA through this stem-loop, the follow-up experiments on the Regnase-1

mRNA 3'UTR represent a logical progression and support the idea that TUT7 level is somehow influencing Regnase-1, directly or indirectly, through its 3'UTR and potentially through a stem-loop structure. The RNA-IP data are not sufficient in themselves to substantiate a direct interaction between TUT7 and the Regnase-1 3'UTR, though they could be consistent with such an interaction.

Our response:

Following the suggestion of the Reviewer #1, we performed RNA-EMSA using different chemically synthesized RNAs to demonstrate the direct interaction between TUT7 and *Zc3h12a* stem-loop. Our results clearly show that TUT7 binds RNA containing *Zc3h12a* stem-loop, but not *Ii6* stem-loop (Fig. 5c). These results further support our model that TUT7 directly interacts with *Zc3h12a* 3' UTR through the stem-loop structure.

The *in vitro* uridylation assay using synthetic mRNA substrates is intriguing and the resulting data (Fig. 5d) could suggest selective uridylation of the Regnase-1 mRNA 3' end by TUT7, though numerous details and control data are missing. Do the synthetic RNA substrates carry a poly(A) tail, as they would *in vivo*, and is it clear that each of the RNA substrates used is capable of 3' end modification by a control nucleotidyl transferase such as poly(A) polymerase?

Our response:

The RNA transcripts generated from *in vitro* transcription do not carry poly(A) tail. We did try to couple *in vitro* transcription and polyadenylation by supplementing poly(A) polymerase into the reaction to generate RNA carrying poly(A) tail. Unfortunately we obtained smeared bands, which we could not get conclusive results. Nevertheless, our RNA-EMSA results in Fig. 5c indicate that TUT7 binding to *Zc3h12a* stem-loop structure is independent of the length of poly(A). In addition, we expended the *in vitro* uridylation analysis using transcripts generated from different combinations of CDS and 3' UTR or the stem-loop structure from *Zc3h12a* or *Ii6*. The results again demonstrate that TUT7 only uridylates transcripts containing *Zc3h12a* 3' UTR or the stem-loop, but not others. These results together suggest that TUT7 selectively uridylates *Zc3h12a* through its stem-loop structure.

Are accessory factors required for the apparent interaction between TUT7 and the Regnase-1 3'UTR, by analogy with TUT4 binding to pre-miRNAs?

Our response:

We did IP TUT7 from LPS-treated macrophages and examined the existence

of LIN28B, a key molecule involved in miRNA uridylation, but did not find their association by immunoblotting. In addition, TUT7 does not interact with Regnase-1 (Supplementary Fig. 8b). Thus, we conclude that neither LIN28B nor Regnase-1 participates in TUT7-mediated *Zc3h12a* uridylation. However, we cannot rule out the possibility of the involvement of additional molecule(s) in TUT7-mediated *Zc3h12a* uridylation.

In summary, I find this an intriguing and potentially valuable study, though the key question of how TLR4 engagement leads to TUT7 induction is not addressed. In addition, the following points in my view render the manuscript unsuitable for publication in its current form.

Our response:

As in our response to the Reviewer #1's comment #2, we investigated the underlying mechanism by which TLR4 induces TUT7, and the results are shown in the revised manuscript (Supplementary Fig. 1a). Our results clearly reveal that only IKK, but not MAPKs JNK nor ERK, is required for TLR4-stimulated TUT7 expression. Extending this investigation to other TLR ligands, we found that IKK, but not JNK, activation is required for TUT7 induction by TLR1/2 and TLR7/8 ligands (Supplementary Fig. 1c). Thus, it could be a general mechanism that induction of TUT7 by TLR ligands is through the IKK/NF- κ B pathway.

1. The 3' RACE data in Figure 6c are deeply puzzling. The cartoon at the top of the figure part suggests that the sequences shown are downstream from the mRNA 3' end – in other words, the cleavage/polyadenylation site, so it would be expected that the sequences would be predominantly poly(A), with perhaps additional untemplated Ts at the 3' end. Instead, many of the sequences do not contain poly(A) tracts; I assume the intention is instead to show for each mRNA the hexanucleotide poly(A) signal, highlighted in yellow, and all sequences downstream from this, in other words the 3' end of the 3' UTR plus any non-templated nucleotides. But the sequence highlighted in yellow is in numerous instances AAATAA (all are in non-inducible transcripts), which unlike the canonical AATAAA would not be expected to act as a functional poly(A) signal. Authentic mRNA sequences would be expected to have 15-30 nt of templated nucleotides downstream from this signal before the cleavage site and poly(A) tail, but this is seen only in a minority of the sequences shown. More worryingly still, the non-poly(A) sequences are mostly different from each other, contain unexplained gaps and do not obviously correspond to the 3' UTR

of Regnase-1. For example, BLAST indicates that the first sequence has no significant match to the murine or human genome, but a highly significant 44/44 match to a number of bacterial cloning vectors; similarly, the single transcript with an untemplated U in the control TUT7+/+ sample shows no significant match to the murine or human genome, but is highly similar to a bacterial 16S rRNA sequence. Given this, it is by no means clear how the final T in this sequence has been designated as 'untemplated'. In my view no conclusions may be drawn from these data. As a consequence, the authors' conclusions about the role of TUT7 in the uridylation of Regnase-1 mRNA, and the relationship between this modification and poly(A) tail length are wholly unsound. Similarly, the first sequence shown in Supplemental Figure 8B seems to be derived not from the IL6 gene, as claimed, but from a bacterial cloning vector. I have not taken the time to identify the remaining sequences, but this is a fairly basic part of the bioinformatic analysis that should have been carried out by the authors. Unfortunately, none of the 3'RACE data presented would appear to be informative.

Our response:

It was our mistake for not carefully confirming the sequences in the original Fig. 6 and Supplementary Fig. 8. As mentioned above, we took the suggestion by reviewer #1 and #3 and performed TAIL-Seq to deep sequence 3' ends of *Zc3h12a* and *Il6*. We excluded genomic-encoded transcripts and transcripts without PAS. The results are shown in Fig. 6 and Supplementary Fig. 9. A previous study revealed that besides canonical polyadenylation sequence (PAS, AAUAAA), which exists in 85-90% of mouse 3'-UTRs, there are 17 types of non-canonical PAS (Gruber AJ et al., 2016, Genome Res., PMID: 27382025). We found canonical PAS in the 3'-UTR of both *Zc3h12a* and *Il6*. The TAIL-Seq results confirm that most transcripts have this canonical PAS and its downstream sequences. We also found a 10-25 nt templated region between PAS and poly(A) tails on 3' end of *Zc3h12a*. Some examples of the sequences are shown in our response to the Reviewer #2 (Comment 1).

2. The immunofluorescence co-localisation data presented in Figure 7 are of marginal value; as both TUT7 and DIS3L2 are both known to be cytoplasmic proteins, their apparent co-localisation adds little additional information.

Our response:

We agree with the reviewer and removed all the figures regarding Dis3L2 from the revised manuscript.

REVIEWER COMMENTS

Reviewer #1 (Remarks to the Author):

The manuscript has been strongly improved, especially due to the inclusion of tail seq experimentation. However, there are several additional points that should be addressed.

1) the labelling of inhibitors in S1a is misleading, since IKKi is another IKK kinase paralog (IKKepsilon)

2) Why do the authors not comment on the obvious effect of p38 inhibitor?

3) the authors should continue language editing: "...111 genes involving in the innate immune response" should be rephrased in "111 genes involved in the innate immune response", also, I find at least 5x a misspelling of Regnase-1 in text and figures (Reganse-1) and 2x Zc3h12a (Zch312a)

4) I still think that the reporter assays in Fig. 2 and 4 are only suggestive for the underlying mechanism, but are not reflecting the endogenous regulation, since the extent of regulation has merely the same tendency but shrinks dramatically in magnitude. The authors should at least mention that the endogenous regulation appears more profound and may therefore involve additional contributions but other factors and mechanisms and discuss their possible identity.

5) Most importantly, the Tut7 RNA-EMSA (Fig. 5c) are not convincing, since the interaction is not strong and the binding reaction is done in the absence (!) of competitor RNA and does not have a control for specificity (ssDNA does not answer this question and neither does the IL6 stem loop, which apparently is on a separate gel and has less RNA or is shown in a lower exposure). The authors should use competitor RNA and compare the binding to the WT, St1m, St2m and St1+2m Zc3h12a stem-loop, since they strongly argue with an importance of these sequence alterations in Fig. 4e. Sequence-independent interactions with RNA by Tut7 would explain the observed independence from oligo-A sequences.

In my view it would not make sense if Tut7 was specific for the binding of some specific stem-loop structure (like histone mRNAs and Zc3h12a, but not IL-6), but rather if it was attracted by other factors (Regnase-1, Roquin-1, Eri-1, Upf1, etc.) to the Zc3h12a mRNA and histone mRNAs as well as many others. The authors should reconsider this possibility and include a discussion the mode of interaction with RNA.

Reviewer #2 (Remarks to the Author):

My earlier comments have been adequately addressed, and the manuscript has been improved.

Reviewer #3 (Remarks to the Author):

The authors have satisfactorily addressed all the points raised in my review of the previous version of their manuscript. In particular, the manuscript is now strengthened by the addition of TAIL-Seq analysis and the removal of all the problematic data from the earlier version.

Point-by-point response to reviewer's comments:

Editor:

You will see that, while the reviewers find that your revisions improved the manuscript, some important points remain to be addressed. Please provide TUT7 EMSA data with additional controls. Please revise your manuscript, addressing all the remaining issues raised by Reviewer #1.

Our response:

We appreciate the opportunity to revise our manuscript. Following Reviewer #1's suggestion, we performed RNA-EMSA with several controls to assess the binding specificity of TUT7 to *Zc3h12a* and *Ii6* stem-loop structures on their 3'-UTRs. The results shown in Fig. 5c-d clearly demonstrate that TUT7 specifically interacts with *Zc3h12a*, but not *Ii6*, stem-loop structure. We also revised our manuscript in response to other issues raised by Reviewer #1 in this revised manuscript.

Reviewer #1:

The manuscript has been strongly improved, especially due to the inclusion of tail seq experimentation. However, there are several additional points that should be addressed.

Our response:

We appreciate the reviewer for his/her helpful comments and suggestions.

1. the labelling of inhibitors in S1a is misleading, since IKKi is another IKK kinase paralog (IKKepsilon).

Our response:

We apologize for the confusion generated from Supplementary Fig. 1a in the last version of our manuscript. We remade the figures with appropriate labeling for inhibitors in supplementary Fig. 1a and c of this revised manuscript to avoid misleading readers.

2. Why do the authors not comment on the obvious effect of p38 inhibitor?

Our response:

Following the reviewer's suggestion, we modify our manuscript to state the effect of p38 inhibitor on LPS-induced TUT7 expression in Results (Page 7, lines 8-10), and discussed its relevance to TUT7 expression by LPS in page 17 (lines 11-19). Basically, our data indicate that TLR4-induced TUT7 expression requires the activities of IKK and, to a lesser extent, p38 MAPK.

These results suggest that their downstream transcription factors might be involved in TUT7 expression triggered by TLR4 activation. We and others have shown that p38 MAPK regulates some LPS-induced genes via several transcription factors, including C/EBP β and CREB. We did identify the conserved binding motif for C/EBP β on both human and murine TUT7 promoters, suggesting that p38 MAPK may modulate LPS-induced TUT7 expression via C/EBP β . However, this notion needs to be confirmed.

3. the authors should continue language editing: "...111 genes involving in the innate immune response" should be rephrased in "111 genes involved in the innate immune response", also, I find at least 5x a misspelling of Regnase-1 in text and figures (Reganse-1) and 2x Zc3h12a (Zch312a).

Our response:

The reviewer's correction is much appreciated. We carefully checked our manuscript and corrected these misspellings. The sentence raised by the reviewer is rephrased as "Among the 352 genes, 111 genes involved in the innate immune response were downregulated in *Tut7*^{-/-} cells." in Page 8, line 11. This revised manuscript was edited again by an English-proficient scientist.

4. I still think that the reporter assays in Fig. 2 and 4 are only suggestive for the underlying mechanism, but are not reflecting the endogenous regulation, since the extent of regulation has merely the same tendency but shrinks dramatically in magnitude. The authors should at least mention that the endogenous regulation appears more profound and may therefore involve additional contributions but other factors and mechanisms and discuss their possible identity.

Our response:

We agree with the reviewer that endogenous regulation of TLR4-triggered inflammation is very complicated and involves many factors and various mechanisms. The reporter assays in Figs. 2 and 4 are a surrogate to monitor gene expression, allowing us to quickly narrow down the possible regulatory region responsible for TUT7-mediated modulation of the expression of IL-6 (Fig. 2) and Zc3h12a (Fig. 4). This approach is a common and simple way to identify the *cis*-element involved in gene regulation.

A paragraph in Discussion (Page 20 line 18 to Page 21 line 6) is rewritten to emphasize that endogenous regulation of inflammatory mediators is controlled by many factors and multiple mechanisms. We also point out in

the last paragraph of Discussion that TUT7-mediated *Zc3h12a* uridylation as a novel posttranscriptional mechanism in regulation of TLR4-driven inflammatory cytokine response (page 21 lines 7-9) to emphasize that our finding is one of multiple mechanisms of regulation of inflammatory response.

5. Most importantly, the Tut7 RNA-EMSA (Fig. 5c) are not convincing, since the interaction is not strong and the binding reaction is done in the absence (!) of competitor RNA and does not have a control for specificity (ssDNA does not answer this question and neither does the IL6 stem loop, which apparently is on a separate gel and has less RNA or is shown in a lower exposure). The authors should use competitor RNA and compare the binding to the WT, St1m, St2m and St1+2m *Zc3h12a* stem-loop, since they strongly argue with an importance of these sequence alterations in Fig. 4e.

Sequence-independent interactions with RNA by Tut7 would explain the observed independence from oligo-A sequences.

In my view it would not make sense if Tut7 was specific for the binding of some specific stem-loop structure (like histone mRNAs and *Zc3h12a*, but not IL-6), but rather if it was attracted by other factors (Regnase-1, Roquin-1, Eri-1, Upf1, etc.) to the *Zc3h12a* mRNA and histone mRNAs as well as many others. The authors should reconsider this possibility and include a discussion the mode of interaction with RNA.

Our response:

The reviewer's suggestion is appreciated. We performed RNA-EMSA and competition assay including RNA oligomers containing *Il6* or *Zc3h12a* WT, St1m, St2m, and St(1+2)m stem-loop mutants. Our results in Fig. 5c show that TUT7 only associates with RNAs containing *Zc3h12a* WT and St(1+2)m stem-loop, but not *Il6* stem-loop, *Zc3h12a* St1m or St2m mutants. In addition, addition of excess RNA oligomers containing *Zc3h12a* WT or St(1+2)m, but not St1m or St2m, stem-loop mutant abolished TUT7 binding to Cy3-labeled RNA probe containing *Zc3h12a* stem-loop (Fig. 5d), suggesting that the interaction of TUT7 and *Zc3h12a* is dependent on its stem-loop structure rather than its sequence. We did discuss the possibility why TUT7 only binds to *Zc3h12a*, but not *Il6*, stem-loop structure in Page 19, line 17 to Page 20, line 4. Eukaryotic histone mRNAs are the only mRNAs that lack poly(A). It is still unclear the molecular mechanism for TUT7 recruitment to histone mRNAs. We therefore did not discuss TUT7 binding to histone mRNAs in our manuscript. Nevertheless, several molecules that cooperate with Regnase-1 to bind to its target mRNAs have been identified. Lin28 is shown to recognize

a GGAG loop and subsequently recruit TUT4 to pre-let-7. We therefore suggest that an unknown factor may be required for TUT7 binding to *Zc3h12a* stem-loop structure in Page 20, lines 6-8.

Reviewer #2:

My earlier comments have been adequately addressed, and the manuscript has been improved.

Reviewer #3:

The authors have satisfactorily addressed all the points raised in my review of the previous version of their manuscript. In particular, the manuscript is now strengthened by the addition of TAIL-Seq analysis and the removal of all the problematic data from the earlier version.

REVIEWERS' COMMENTS

Reviewer #1 (Remarks to the Author):

The authors have addressed my remaining concerns. The EMSA results are very clear now, I think it was worth the extra effort. Congratulations.